# miRNA-mediated gene silencing in *Drosophila* larval development involves GW182-dependent and independent mechanisms

Eriko Matsuura-Suzuki[1,2], Kaori Kiyokawa[1,3], Shintaro Iwasaki [ID][2,4] & Yukihide Tomari [ID][1,4][✉]

## Abstract

**MicroRNAs (miRNAs) regulate a wide variety of biological processes by silencing their target genes. Argonaute (AGO) proteins load miRNAs to form an RNA-induced silencing complex (RISC), which mediates translational repression and/or mRNA decay of the targets. A scaffold protein called GW182 directly binds AGO and the CCR4-NOT deadenylase complex, initiating the mRNA decay reaction. Although previous studies have demonstrated the critical role of GW182 in cultured cells as well as in cell-free systems, its biological significance in living organisms remains poorly explored, especially in *Drosophila melanogaster*. Here, we generated *gw182*-null flies using the CRISPR/Cas9 system and found that, unexpectedly, they can survive until an early second-instar larval stage. Moreover, in vivo miRNA reporters can be effectively repressed in *gw182*-null first-instar larvae. Nevertheless, *gw182*-null flies have defects in the expression of chitin-related genes and the formation of the larval trachea system, preventing them from completing larval development. Our results highlight the importance of both GW182-dependent and -independent silencing mechanisms in vivo.**

**Keywords** GW182; MicroRNA; Gene Silencing; Development; *Drosophila melanogaster*
**Subject Categories** Development; RNA Biology

## Introduction

MicroRNAs (miRNAs) are small non-coding RNAs of ~22 nucleotides (nt) in length, which silence the expression of target genes and regulate a wide variety of biological processes. miRNAs are loaded into a member of the Argonaute (AGO) protein family to form the effector ribonucleoprotein complex called RNA-induced silencing complex (RISC). miRNA-loaded RISC binds to complementary target sites, typically located within the 3′ untranslated region (3′ UTR) of mRNAs, and induces translational

repression and/or mRNA deadenylation/decay (Bartel, 2018; Iwakawa and Tomari, 2022). In *Drosophila melanogaster*, miRNAs mainly function through Ago1-containing RISC, whereas small interfering RNAs (siRNAs), another class of small RNAs crucial for silencing of viruses and transposons, function through Ago2-containing RISC (Ghildiyal et al, 2008; van Rij et al, 2006; Kawamura et al, 2008).

Many AGO proteins associate with GW182 as a partner protein, which plays an important role in miRNA-mediated gene silencing. The N-terminal glycine/tryptophan (GW) repeats of GW182 directly binds the three hydrophobic pockets (tryptophane-binding pockets) in the PIWI domain of AGO proteins (Schirle et al, 2014; Elkayam et al, 2017). GW182 further recruits the CCR4-NOT deadenylase complex and the PAN2-PAN3 deadenylase complex, mainly through its C-terminal region called the "silencing domain" (Jonas and Izaurralde, 2015). Agreeing with the role of GW182 as the scaffold between AGO and the deadenylase complexes, previous studies have demonstrated the essential requirement of GW182 in miRNA-mediated deadenylation and decay of target mRNAs (Braun et al, 2011; Chekulaeva et al, 2011; Fabian et al, 2011; Fukaya and Tomari, 2011). On the other hand, translational repression by miRNAs is reported to occur in two different—GW182-dependent and -independent—manners, at least in cultured cells and in cell-free systems (Fukaya and Tomari, 2012; Fukaya et al, 2014; Fukao et al, 2014; Jonas and Izaurralde, 2015; Kuzuoğlu-Öztürk et al, 2016). However, the physiological importance of GW182 and its modes of action in vivo remain poorly explored.

Multiple GW182 paralogs (TNRC6A, B, and C in humans) exist in many species. *Caenorhabditis elegans* have two GW182 paralogs, AIN-1 and AIN-2, but only AIN-1 can associate with the deadenylase complexes (Ding et al, 2005; Zhang et al, 2007; Kuzuoglu-Öztürk et al, 2012). Importantly, loss of both AIN genes does not cause significant lethality at the embryonic stage, although it later leads to a larval developmental arrest (Jannot et al, 2016). Moreover, the embryonic lethality caused by the loss of ALG-1 and ALG-2, two miRNA-specific AGOs in *C. elegans*, can be rescued by a tryptophan-binding pocket mutant of ALG-1, which cannot associate with AIN-1 or AIN-2 (Jannot et al, 2016). Moreover, in the germline of *C. elegans*, AIN-1 is dispensable but instead the

[1]Institute for Quantitative Biosciences, The University of Tokyo, Bunkyo-ku, Tokyo 113-0032, Japan. [2]RNA Systems Biochemistry Laboratory, RIKEN Cluster for Pioneering Research, Wako, Saitama 351-0198, Japan. [3]Institute of Industrial Science, The University of Tokyo, Meguro-ku, Tokyo 153-8505, Japan. [4]Department of Computational Biology and Medical Sciences, Graduate School of Frontier Sciences, The University of Tokyo, Bunkyo-ku, Tokyo 113-0032, Japan. ✉E-mail: tomari@iqb.u-tokyo.ac.jp

DEAD-box helicase GLH-1 (a homolog of fly Vasa) is required for miRNA-mediated gene regulation (Dallaire et al, 2018). These studies strongly suggest that GW182 is not essential for some biological contexts in *C. elegans*. In contrast to worms, *Drosophila melanogaster* has only one GW182 gene called *gawky* (*gw*). Gawky protein (called fly GW182 thereafter) is 1,384 aa in length and binds Ago1, the miRNA-specific AGO in flies. *gw182* gene locates in the tiny 4th chromosome, which lacks crossovers during meiosis and is thus difficult to genetically manipulate. Indeed, *gw¹* is the only mutant fly allele reported in the literature so far (Schneider et al, 2006). *gw¹* expresses a truncated GW182 protein (1–966 aa), which is also detectable in healthy heterozygous adults and thus does not seem to have dominant-negative effects. Yet, homozygous *gw¹* mutant embryos fail to cellularize and die before gastrulation. This phenotype of *gw¹* is, oddly, even more severe than that of a null allele of *ago1*, which becomes lethal only after segment formation of embryos (Kataoka et al, 2001). These observations raise the possibility that the specific truncated form of GW182 and/ or additional mutation(s) in the *gw¹* allele causes the secondary effect, highlighting the necessity of generating and characterizing bona fide *gw182*-null flies to understand its physiological role.

Here, we utilized the CRISPR-Cas9 system and successfully knocked out the *gw182* gene in the challenging 4th chromosome. We found that, unlike *gw¹*, *gw182*-null flies can survive until an early point of the second-instar larval stage. Moreover, in vivo reporters can be effectively silenced in *gw182*-null first-instar larvae. Nonetheless, our RNA-seq analysis revealed that *gw182*-null flies have defects in the expression of chitin-related genes and the formation of the larval trachea system, explaining their failure to complete the larval development. Our results highlight the in vivo importance of both GW182-dependent and -independent silencing mechanisms by miRNAs.

# Results

## Development of an in vivo reporter that can monitor the miR-8 activity in flies

To accurately monitor the silencing activity of fly miRNAs in vivo, we set up a new reporter system with multiple controls. We designed a set of fluorescent reporters under the control of the GAL4-UAS system, a general method to activate gene expression in a tissue- and/or development-specific manner in flies (Brand and Perrimon, 1993). The first GFP reporter includes two copies of a typical, centrally bulged miR-8 target site in the 3′ UTR (the Sensor reporter; Fig. 1A,B). As a negative control, the second GFP reporter harbors two copies of mutated miR-8 sites with mismatches in the "seed" region, which is critical for miRNA target recognition (the Mutated reporter; Fig. 1A,B). Together with either the Sensor or Mutated GFP reporter, we co-expressed an independent mCherry reporter without miRNA target sites as an internal control and normalized the GFP expression by the mCherry expression (Fig. 1A).

We expressed these reporters using an eye-specific driver, GMR-GAL4, with or without overexpression of miR-8 (Fig. 1C,D). Since endogenous miR-8 is not expressed in the eye disc at the third instar larval stage, the GFP fluorescence of the Sensor reporter remained unsilenced and was readily detectable in the eye disc

(Fig. 1C,D). Upon miR-8 overexpression, however, the GFP intensity of the Sensor reporter was significantly downregulated, while the mCherry fluorescence of the internal control reporter was unaffected (Fig. 1C,D). In contrast to the Sensor reporter, the GFP fluorescence of the Mutated reporter was expectedly observed regardless of the overexpression of miR-8 (Fig. 1C,D). miR-8-specific silencing of the Sensor GFP reporter, but not the Mutated reporter, was also confirmed at the mRNA level (Fig. 1E). These results demonstrate that this reporter system can accurately monitor the miR-8 silencing activity in living flies.

## Ago1-RISC can repress the miR-8 reporter independently of GW182 in the eye disc

We next investigated the impact of GW182 on miRNA-mediated silencing in vivo. We knocked down *gw182* or *ago1* using corresponding inverted repeats (IRs) as the source of dsRNAs to trigger RNA interference (RNAi) (Dietzl et al, 2007; Perkins et al, 2015) and evaluated the expression levels of the Sensor reporter or the Mutated reporter in the presence or absence of miR-8 in the eye disc of the third instar larvae. Quantitative RT-PCR (qRT-PCR) - based quantification of *gw182* or *ago1* mRNA from the dissected eye disc showed only a modest decrease by the corresponding RNAi (Fig. EV1A,B), presumably due to inevitable contamination of the surrounding tissues. Nevertheless, both *gw182* and *ago1* RNAi in the eye disc caused a rough eye phenotype in adult flies associated with miRNA dysfunction, suggesting the knockdown was functionally effective (Fig. EV1C). As expected, depletion of *ago1*, but not depletion of *ago2*, abrogated miR-8-dependent silencing of the Sensor reporter both at the protein level (Figs. 2A,C and EV2A,B) and mRNA level (Figs. 2D and EV2C), confirming that miR-8 acts through Ago1-RISC. In contrast, the Sensor reporter remained silenced upon knockdown of *gw182* using two independent IRs (Figs. 2A,C,D and EV2). As a control, the Mutated reporter was insensitive to *ago1* or *gw182* knockdown (Figs. 2B–D and EV2). These results raise the possibility that AGO1-RISC can silence miRNA targets independently of GW182 in the fly eye disc.

## GW182 is required for larval development only after an early second-instar stage

In general, residual proteins can remain upon IR-mediated knockdown, and it is difficult to exclude their contribution (Fig. EV1A,B). Therefore, we sought to generate *gw182*-null alleles using the CRISPR-Cas9 system (Gratz et al, 2013) and designed a single guide RNA (sgRNA) that targets the first exon of the *gw182* gene (Fig. 3A). Despite its challenging genomic location in the tiny 4th chromosome, we successfully obtained two independent mutant alleles (called *gw^null-1^* and *gw^null-2^* thereafter). These alleles had a small deletion that leads to a premature stop codon in the coding sequence of *gw182* (Fig. 3B,C), which is expected to block the translation of all known functional domains of GW182 and cause nonsense-mediated mRNA decay. Unlike the previously reported phenotype of *gw¹*, homozygous embryos of *gw^null-1^* or *gw^null-2^* completed embryogenesis and hatched normally (Fig. 3D). However, most of *gw^null-1^* or *gw^null-2^* larvae died after molting into second instars, at ~50 h after egg laying (Fig. 3D). It has been demonstrated that maternally deposited GW182 starts to decline at 60–70 min after embryo laying (Schneider et al, 2006). Although the fate of maternally deposited GW182 after 60–70 min in

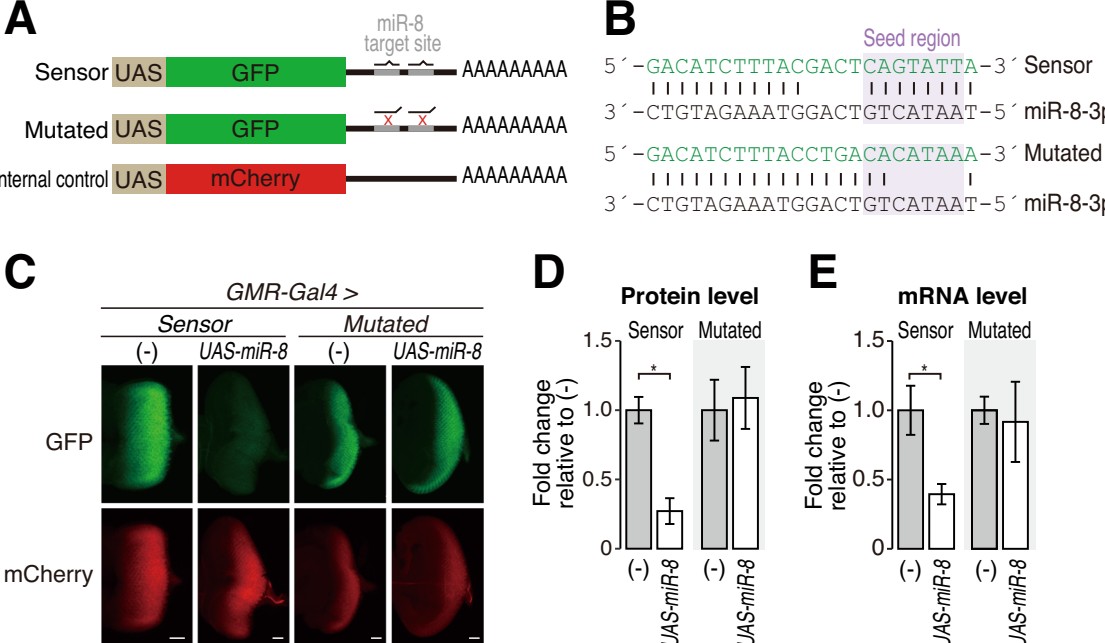

**Figure 1. Construction and validation of the miR-8 reporter in eye discs.**

(A) Schematic diagrams of reporter constructs bearing GFP with two miR-8 target sites (top) or two seed-mismatched sites (middle), and mCherry without miR-8 target sites as an internal control (bottom). (B) Schematic representation of the pairing between miR-8 and the Sensor reporter or the Mutated reporter for each miRNA-binding site. Lines represent base pairs. (C) Fluorescence microscopy of the Sensor reporter or the Mutated reporter in the eye disc of third instar larvae. GFP (green) and mCherry (red) expressions were compared with or without miR-8 overexpression. Scale bar: 20 μm. (D) Quantification of the relative GFP protein level in (C). The GFP fluorescence levels were normalized to the mCherry fluorescence levels, and the fold changes are shown relative to the average value without miR-8 overexpression as mean ± SD. Error bars represent standard derivation from independent eye discs for each sample. $P$ values were determined by Student's $t$ test (unpaired, two-sided). *$P = 3.46$e-8. (E) Quantification of the relative GFP mRNA level by qRT-PCR for (C). The GFP mRNA levels were normalized to the mCherry mRNA levels, and the fold changes are shown relative to the value without miR-8 overexpression as mean ± SD. Error bars represent standard derivation from three independent experiments. $P$ values were determined by Student's $t$ test (unpaired, two-sided). *$P = 5.37$e-3. Genotypes (C–E): *GMR-GAL4/UAS-Sensor reporter; +/+* ($n = 8$), *GMR-GAL4/UAS-Sensor reporter; UAS-miR-8/+* ($n = 5$), *GMR-GAL4/UAS-Mutated reporter;+/+* ($n = 10$), and *GMR-GAL4/UAS-Mutated reporter;UAS-miR-8/+* ($n = 10$). The numbers ($n$) are for (D). Source data are available online for this figure.

unknown, the time point of death of *gw182*-null flies is more than 48 h after this time window. In the surviving first-instar larvae of *gw^{null-1}* and *gw^{null-2}*, only trace amounts of GW182 protein were detectable (Fig. 3E). These results suggest that flies can survive embryogenesis and first-instar larval development even after maternal GW182 is mostly depleted, but GW182 becomes essential for development at an early second-instar stage.

We then performed a series of rescue experiments using *gw^{null-2}*, which has a larger genomic deletion and is thus more convenient for genotyping than *gw^{null-1}*. The developmental arrest in *gw^{null-2}* was fully rescued by the expression of full-length GW182 driven by the endogenous promoter of *gw182*. A GW182 fragment (1–605), which lacks the entire silencing domain, could partially rescue the lethality at the second-instar larval stage (Fig. 3F), agreeing with a previous notion that functional domains of fly GW182 are not exclusively localized to the silencing domain but rather dispersed throughout the protein (Fukaya and Tomari, 2011). A longer GW182 fragment (1–940 aa), which is shorter than the truncated GW182 expressed in *gw^1* (1–966 aa) but was previously shown to largely retain the silencing activity in a cell-free system (Fukaya and Tomari, 2011), could rescue the lethality not only at the second-instar stage but also partly at the third-instar stage (Fig. 3F). These results together imply that the severe

embryonic lethal phenotype previously reported for *gw^1* (Schneider et al, 2006) is likely caused by a secondary mutation(s) in the genome (also see "Discussion").

## GW182 is not required for silencing the miR-8 reporter at the first-instar stage

To confirm that miRNAs can silence the miR-8 reporter in the absence of GW182 (Fig. 2), we crossed *gw^{null-2}* with the reporter strains and monitored their expression levels at the first-instar larval stage before they die (Fig. 3D). Because miR-8 is highly expressed from embryo to larval stages (Ruby et al, 2007), we expressed the Sensor reporter or the Mutated reporter using the promotor sequence of miR-8. The Sensor reporter, but not the Mutated reporter, is thus repressed by endogenous miR-8 in this assay (Fig. EV3). We found that repression of the Sensor reporter relative to the Mutated reporter was comparable between *gw^{null-2}*–/– homozygotes and *gw^{null-2}*–/+ heterozygotes, both at the protein (Fig. 4A,B) and mRNA (Fig. 4C) levels. We concluded that miR-8 can silence the in vivo reporter independently of GW182 in first-instar larvae, although it is formally possible that the residual amount of GW182 in *gw^{null-2}* is sufficient to support the full silencing activity.

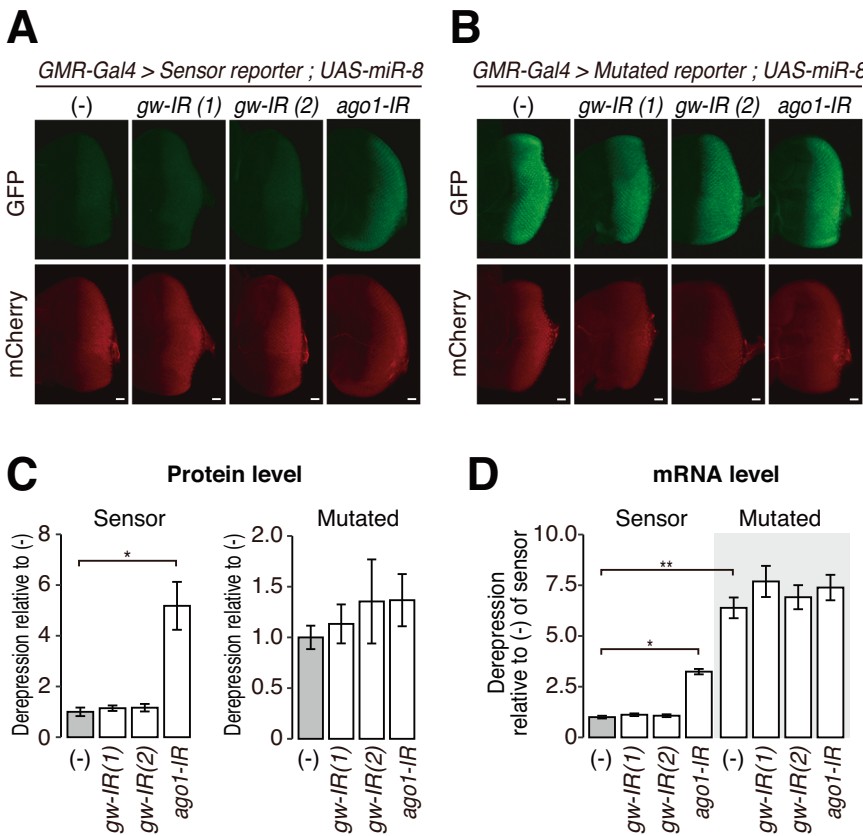

**Figure 2.  GW182 is not essential for miR-8-mediated silencing of the reporter in eye discs.**

(A, B) Fluorescence microscopy images for the Sensor reporter (A) and the Mutated reporter (B) upon miR-8 overexpression. *gw-IR(1)* and *gw-IR(2)* are independent IR-expressing strains, and the indicated genes knock down in the third instar larval eye disc. Scale bar: 20 μm. (C) Quantified derepression at the protein level in (A, B). The GFP fluorescence level was normalized to the mCherry fluorescence level, and the derepression levels are shown relative to the negative control without IR expression (–) of the Sensor reporter or the Mutated reporter as mean ± SD. Note that eye discs in (A, B) were observed with different laser intensities, and the data were normalized to (–) independently. Error bars represent standard derivation from ten independent eye discs for each sample. *P* values were calculated by one-way ANOVA with the post hoc Tukey HSD test. *$P < 1.00e-7$. (D) Quantified derepression at the mRNA level in the Sensor and Mutated reporters. GFP mRNA level was normalized to mCherry mRNA level, and the derepression levels are shown relative to the negative control (–) of the Senor reporter as mean ± SD. Error bars represent standard derivation from three independent experiments. *P* values were calculated by one-way ANOVA with the post hoc Tukey HSD test. *$P = 3.30e-4$, **$P < 1.00e-7$. Genotypes for (A): *GMR-GAL4/ UAS-Sensor reporter; UAS-miR-8/+* ($n = 10$), *GMR-GAL4, GW182-IR(1)/UAS-Sensor reporter;UAS-miR-8/+* ($n = 10$), *GMR-GAL4,GMR-IR(2)/UAS-Sensor reporter; UAS-miR-8/+* ($n = 10$), *GMR-GAL4,ago1-IR/UAS-Sensor reporter; UAS-miR-8/+* ($n = 10$). The numbers ($n$) are for (C). Genotypes for (B): *GMR-GAL4/UAS-Mutated reporter; UAS-miR-8/+* ($n = 10$), *GMR-GAL4,GW182-IR(1)/UAS-Mutated reporter; UAS-miR-8/+* ($n = 10$), *GMR-GAL4,GMR-IR(2)/UAS-Mutated reporter; UAS-miR-8/+* ($n = 10$), and *GMR-GAL4,ago1-IR/ UAS-Mutated reporter; UAS-miR-8/+* ($n = 10$). The numbers ($n$) are for (C). Source data are available online for this figure.

## GW182 is required for the repression of some endogenous miRNA targets

Although repression of the miR-8 reporter did not apparently require GW182 in first-instar larvae (Fig. 4), *gw182*-null flies stopped growing and died after molting into second-instar larvae (Fig. 3D). Thus, it is expected that some endogenous genes are misregulated in the absence of GW182. To explore this possibility, we performed RNA sequencing (RNA-seq) of *gw^{null-2}* homozygous larvae at the first-instar stage, right before they show developmental arrest, using *yw* larvae as a control. Because miR-8 and miR-14 are the two most abundant miRNAs in fly larvae (Ruby et al, 2007), we focused our transcriptome analysis on mRNAs bearing predicted target sites for miR-8 and miR-14. As expected, miR-8 and miR-14 targets were slightly but significantly upregulated in *gw^{null-2}* flies, compared to mRNAs that lack any predicted target sites for the top 10 abundant larval miRNAs (Fig. 5A).

Among all upregulated transcripts in *gw^{null-2}* flies, the predicted targets for miR-8 and miR-14 and those for the top 10 miRNAs accounted for ~15% and ~25%, respectively (Fig. 5B). The expression levels of miR-8 and miR-14 themselves were comparable or even slightly higher in *gw^{null-2}* than in *yw* (Fig. 5C), indicating that the upregulation of miR-8 and miR-14 targets is due to defects in target repression, rather than the absence of those miRNAs. Similar results were obtained in the comparisons between *gw^{null-2}*–/– and –/+ siblings and between *gw^{null-2}*–/– and +/+ siblings (Fig. EV5). Thus, GW182 is required for the proper regulation of at least some miRNA targets.

## GW182 is required for the regulation of chitin-related genes and the formation of the larval trachea system

To further investigate the biological role of fly GW182 in vivo, we focused our RNA-seq analysis on protein-coding genes that are

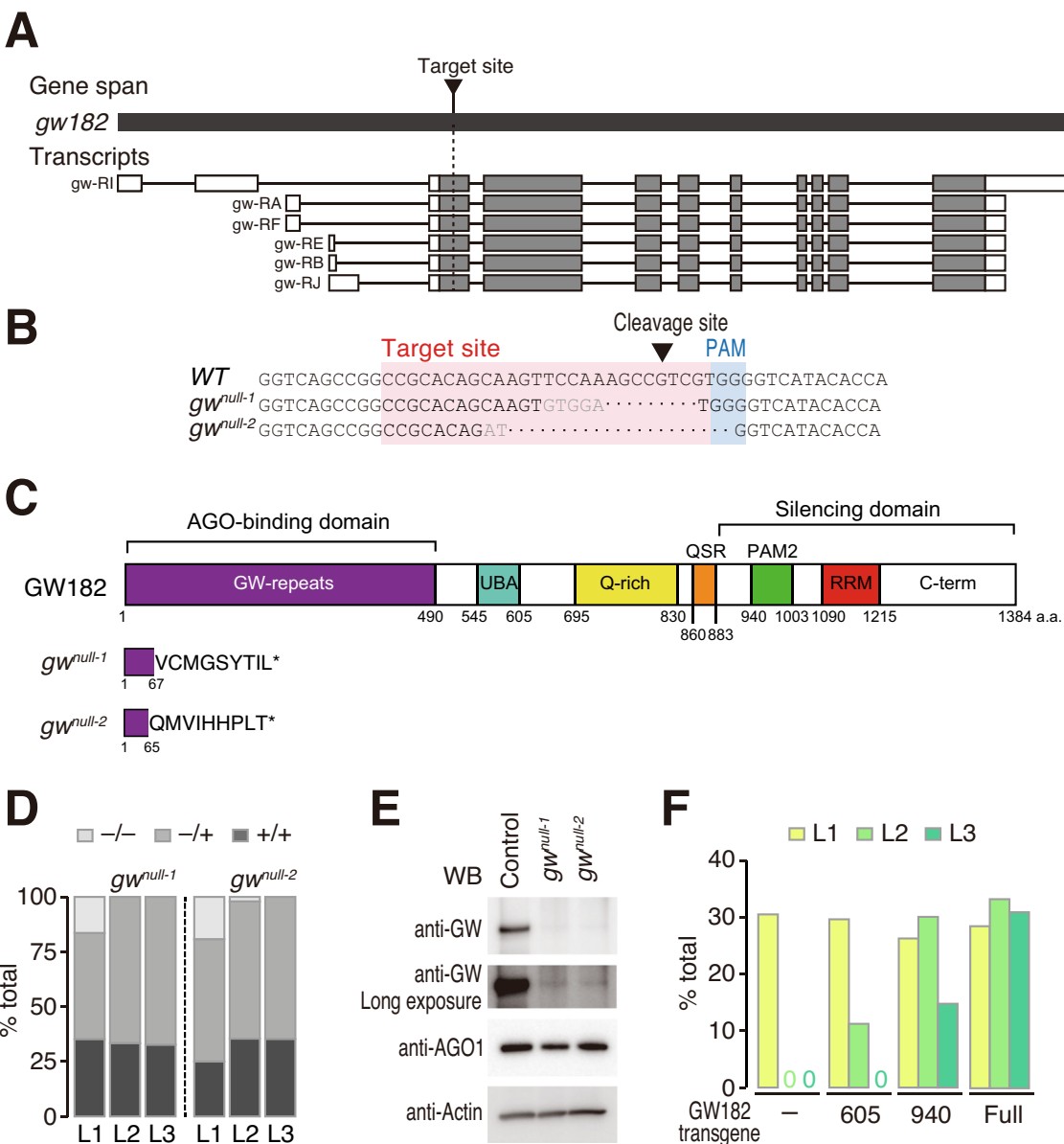

**Figure 3. Generation of *gw182*-null alleles by CRISPR/Cas9 system.**

(**A**) A schematic representation of the *gw182* gene and their transcripts showing with the sgRNA target site. Exons are shown as gray boxes, 5'UTR, and 3'UTR as white boxes. (**B**) sgRNA target sequence and modified sequences in *gw182*-null alleles. Target site and PAM motif are shown in a pink box and blue box, respectively. The cleaved site by Cas9 is indicated by a black arrowhead. (**C**) Domain structure of GW182 protein and mutant proteins expressed in *gw182*-null mutants. (**D**) Genotype ratio at each larval stage of the F1 generation of crossing with heterozygotes of *gw182*-null mutants. (**E**) Western blots of GW182 and AGO1 in homozygotes of *gw182*-null mutant at first-instar larval stage. β-Actin is used as an internal control. (**F**) Percentage of rescued *gw182*-null mutant by GW182 truncated transgenes. The total numbers of individual larvae analyzed for each genotype were: negative control (L1: 95, L2: 85, L3: 57), 605 (L1: 118, L2: 89, L3: 136), 940 (L1: 156, L2: 113, L3: 88), and Full (L1: 84, L2: 156, L3: 116). Source data are available online for this figure.

differently expressed between *yw* and *gw182*-null mutants at the first-instar larval stage. There were many misregulated genes in the two *gw182*-null flies compared to *yw* (Fig. 6A–D), while the gene expression profiles were highly similar in *gw^null-1^* and *gw^null-2^* flies as expected (Fig. 6E). Using stringent criteria (log$_2$FC [fold change] > 1 or < −1; FDR [false discovery rate] <0.05), we found that 358 and 575 genes were enriched or depleted in *gw^null-1^* (Fig. 6A,B) and 480 and 612 genes in *gw^null-2^*, respectively (Fig. 6C,D). Of these, 292 enriched genes

and 386 depleted genes were shared by *gw^null-1^* and *gw^null-2^*. Gene Ontology (GO) analysis of those commonly misregulated genes showed that genes upregulated in *gw182*-null larvae are associated with defense response to Gram-positive bacterium, peptidoglycan metabolic, peptidoglycan catabolic, and glycosaminoglycan catabolic processes (Figs. 6F and EV4A), suggesting that the immune system of *gw182*-null mutants is more sensitive than that of *yw*. Since fly larvae are typically cultured on food that contains raw yeast, a

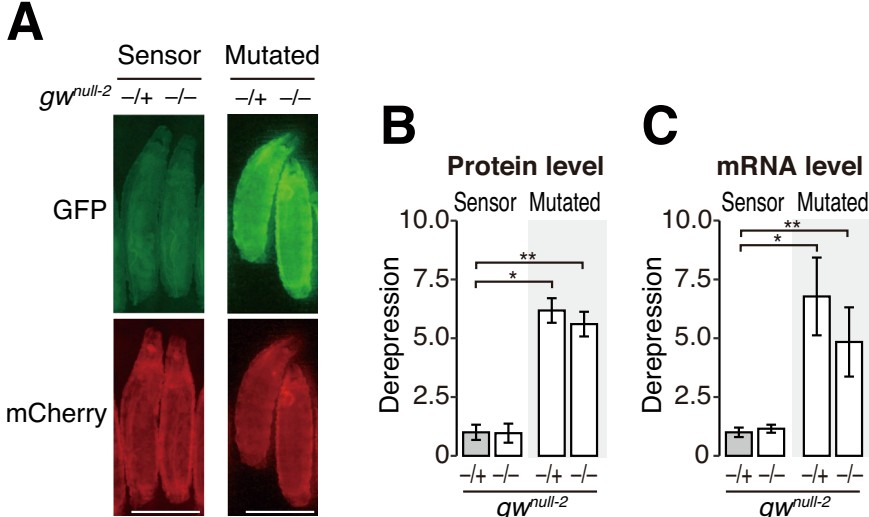

**Figure 4. GW182 are not essential for miRNA-mediated silencing of the sensor reporter at the first-instar larval stage.**

(A) Fluorescence microscopy of the Sensor reporter and the Mutated reporter at first-instar larval stage. GFP (green) or mCherry (red) expressions are compared between heterozygotes (−/+) or homozygote (−/−) of $gw^{null-2}$. Scale bar: 250 μm. (B) Quantification of the derepression level for protein in (A). The fluorescence GFP level of the Sensor reporter or the Mutated reporter was normalized to the fluorescence mCherry level and the derepression level is shown relative to $gw^{null-2}$ −/+ of the Sensor reporter as mean ± SD. Error bars represent standard derivation from the number of animals scored (*N*; see the "Methods" section). *P* values were calculated by one-way ANOVA with the post hoc Tukey HSD test. *$P < 1.00e-7$; **$P < 1.00e-7$. (C) Quantification of the derepression level for mRNA. The GFP mRNA level of the Sensor reporter or the Mutated reporter wes normalized to the mCherry mRNA level, and the depression level is shown relative to the $gw^{null-2}$ −/+ of sensor reporter as mean ± SD. Error bars represent standard derivation from the number of animals scored (*n*; see "Methods"). *P* values were calculated by one-way ANOVA with the post hoc Tukey HSD test. *$P < 1.00e-7$; **$P < 1.00e-7$. Genotypes: *miR-8-GAL4/UAS-Sensor reporter; +/+; $gw^{null-2}$/+* (*n* = 42), *miR-8-GAL4/UAS-Sensor reporter; +/+; $gw^{null-2}$/$gw^{null-2}$* (*n* = 10), *miR-8-GAL4/UAS-Mutated reporter; +/+; $gw^{null-2}$/+* (*n* = 47), and *miR-8-GAL4/UAS-Mutated reporter; +/+; $gw^{null-2}$/$gw^{null-2}$* (*n* = 13). The numbers (*n*) are for (B). Source data are available online for this figure.

Gram-positive fungus (Bianchi, 1965), we first predicted that excess immune response to yeast may be responsible for the lethality of *gw182*-null flies. However, when we cultured *gw182*-null larvae on yeast-free food, they still showed developmental arrest at the early second-instar stage (EM, KK, YT, unpublished observations). Therefore, we next focused on the GO terms of downregulated genes in *gw182*-null flies and detected strong enrichment of chitin-based cuticle development and cuticle development genes (Figs. 6F and EV4B). It is well known that chitin microfibril is required for the formation of body cuticle, trachea epithelial tube, and gut epithelium, protecting flies from invading pathogens and other harmful stresses (Moussian, 2010). Moreover, downregulated genes in *gw182*-null mutants included *vermiform* (*verm*), *serpentine* (*serp*) and *mummy* (*mmy*) (Fig. 6A,C), whose strong genetic associations with the formation of trachea epithelial tube have been demonstrated (Luschnig et al, 2006; Wang et al, 2006; Tonning et al, 2006). We thus examined the trachea trunks of first-instar larvae of $gw^{null-2}$ and *yw* under microscopy. In *yw* larvae, we could clearly observe two thick trachea trunks with air-filled respiratory tubules (Fig. 7A). In contrast, the trachea trunks of $gw^{null-2}$ larvae were overall thin and showed frequent air-filling defects (Fig. 7B). These results indicate that GW182 regulates trachea formation during larval development.

## Discussion

In this study, we utilized the CRISPR/Cas9 system and successfully generated *gw182*-null flies. We found that *gw182*-null flies can

complete the embryogenesis and develop into second-instar larvae. GW182 protein as well as *gw182* mRNA is known to be maternally deposited, which starts to decline at 60–70 min after egg deposition (Schneider et al, 2006). Therefore, GW182 may well be required for very early embryogenesis in flies. Nevertheless, it is remarkable that *gw182*-null first-instar larvae, in which expression of GW182 is only marginally detectable (Fig. 3E), can survive and molt into second instars. This phenotype is much milder than *ago1* null flies (Kataoka et al, 2001), implying the existence of an Ago1-dependent but GW182-independent silencing mechanism by miRNAs in vivo, as reported in *C. elegans* (Jannot et al, 2016). Indeed, we found that the in vivo miR-8 reporter can be effectively silenced in *gw182*-null first-instar larvae (Fig. 4) as well as in the GW182-depleted eye disc of third instar larvae (Fig. 1). On the other hand, *gw182*-null flies showed developmental arrest at an early second-instar stage with misregulation of some miRNA targets (Figs. 3D and 5A), suggesting that a GW182-dependent silencing mechanism is essential for survival after this developmental stage. It has been shown that, in *C. elegans*, somatic miRNAs are dependent on GW182 (AIN-1), whereas germline miRNAs function through another factor, GLH-1 (Dallaire et al, 2018). Although additional studies are required, it is tempting to speculate that *Drosophila* also utilize a similar tissue-specific or developmental stage-specific dependency on GW182 for miRNA-mediated gene regulation.

It was previously reported that $gw^1$ homozygous mutants, which express the 1–966 aa fragment of GW182, fail to cellularize at early embryogenesis (Schneider et al, 2006). This phenotype of $gw^1$ is severer than that of *gw182*-null flies (Fig. 3D). Moreover, we found

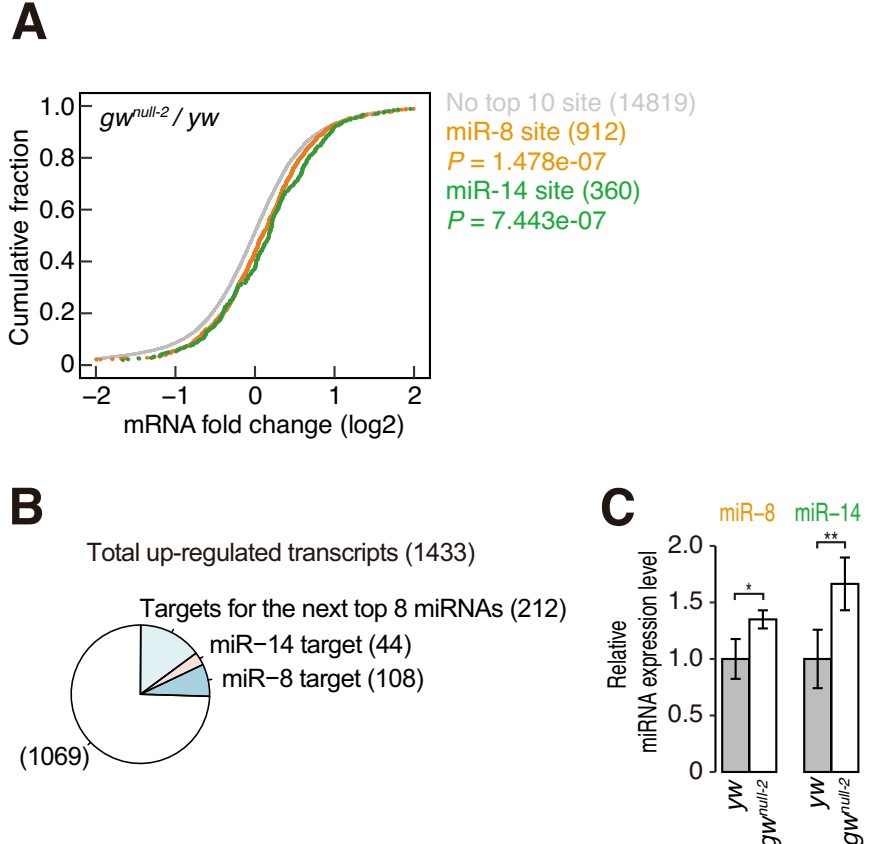

**Figure 5. Depletion of GW182 impaired miRNA-mediated silencing for endogenous miRNA targets at the first-instar larval stage.**

(A) Upregulation of miRNA targets by knockdown of *gw182*. First larvae of *gw182*-null homozygote or *yw* were collected, and their transcriptomes were analyzed by RNA-seq. The fold change values, *gw*$^{null-2}$/*yw*, were calculated for each mRNA. The cumulative fractions of the average fold change values of two independent experiments are shown for the following categories: mRNAs without a target site of the top 10 most abundant miRNAs (gray) and mRNAs with predicted target sites for miR-8 (orange) or miR-14 (green), together with their *P* values of the Mann–Whitney *U* test. (B) Pie chart showing the percentage of predicted miRNA targets within the transcripts upregulated by *gw182* deletion. The numbers of transcripts in each category are shown in parentheses. (C) Quantification of microRNA expression. The levels of miR-8 and miR-14 were measured by qRT-PCR from first-instar larvae of *gw*$^{null-2}$ homozygote or *yw*. *rp49* was used as a reference. The data were normalized by the miRNA expression level of *yw*, shown as mean ± SD. Error bars represent standard derivation from three independent experiments. *P* values were calculated by Student's *t* test (unpaired, two-sided). *$P = 0.035$; **$P = 0.029$. Source data are available online for this figure.

that the developmental arrest of *gw182*-null flies at the third instar larvae can be rescued not only by the full-length GW182 but also largely by the 1–940 aa fragment of GW182 (Fig. 3F). These observations imply that the phenotype of *gw*$^1$ is caused by a secondary mutation(s). The *gw*$^1$ mutant has two identified nucleotide changes in the genome: W967stop in *gw* and N144I in the N-terminal region of *myoglianin (myo)*. Myo is a homolog of vertebrate Myostatin (Mstn/GDf8), a member of transforming growth factor (TGF) family (Hinck et al, 2016). Mstn circulates in blood as a latent complex with the N-terminal prodomain associated with the ligand domain, and proteolytic cleavage is required to release the active ligand domain from the N-terminal prodomain. Since N144 is placed in the N-terminal prodomain of Myo, the N144I mutation is believed not to affect the function of mature Myo. However, it is possible that the mutation blocks the proteolytic cleavage and interferes with its function, which may explain the severe phenotype of *gw*$^1$. Further investigations are needed to clarify this point.

Our RNA-seq analysis suggested that chitin-related genes such as *verm*, *serp*, and *mmy* are all downregulated in the first-instar larvae of *gw*$^{null-1}$ *and gw*$^{null-2}$ (Fig. 6A,C). Chitin is an essential component of larval epidermis, gut, and trachea (Lehane, 1997; Luschnig et al, 2006; Wang et al, 2006; Hegedus et al, 2009). Indeed, *gw182*-null mutants showed defective trachea structures in first-instar larvae (Fig. 7), demonstrating the importance of GW182 in trachea formation. However, those chitin-related genes do not seem to be direct targets of miRNAs, because miRNA target genes are generally upregulated by the loss of GW182. On the other hand, *ribbon (rib)*, a BTB family transcriptional activator/repressor (Silva et al, 2016), was upregulated in *gw182*-null mutants (Fig. 6A,C). Rib is considered as a regulating factor of *verm* and *serp*, and it has been demonstrated that proper trachea formation requires Rib (Bradley and Andrew, 2001; Shim et al, 2001; Luschnig et al, 2006), suggesting a possibility that *verm* and *serp* are misregulated via *rib* in *gw182*-null flies. Future studies are warranted to clarify the relationship between miRNA targets and their associated genes that regulate trachea formation.

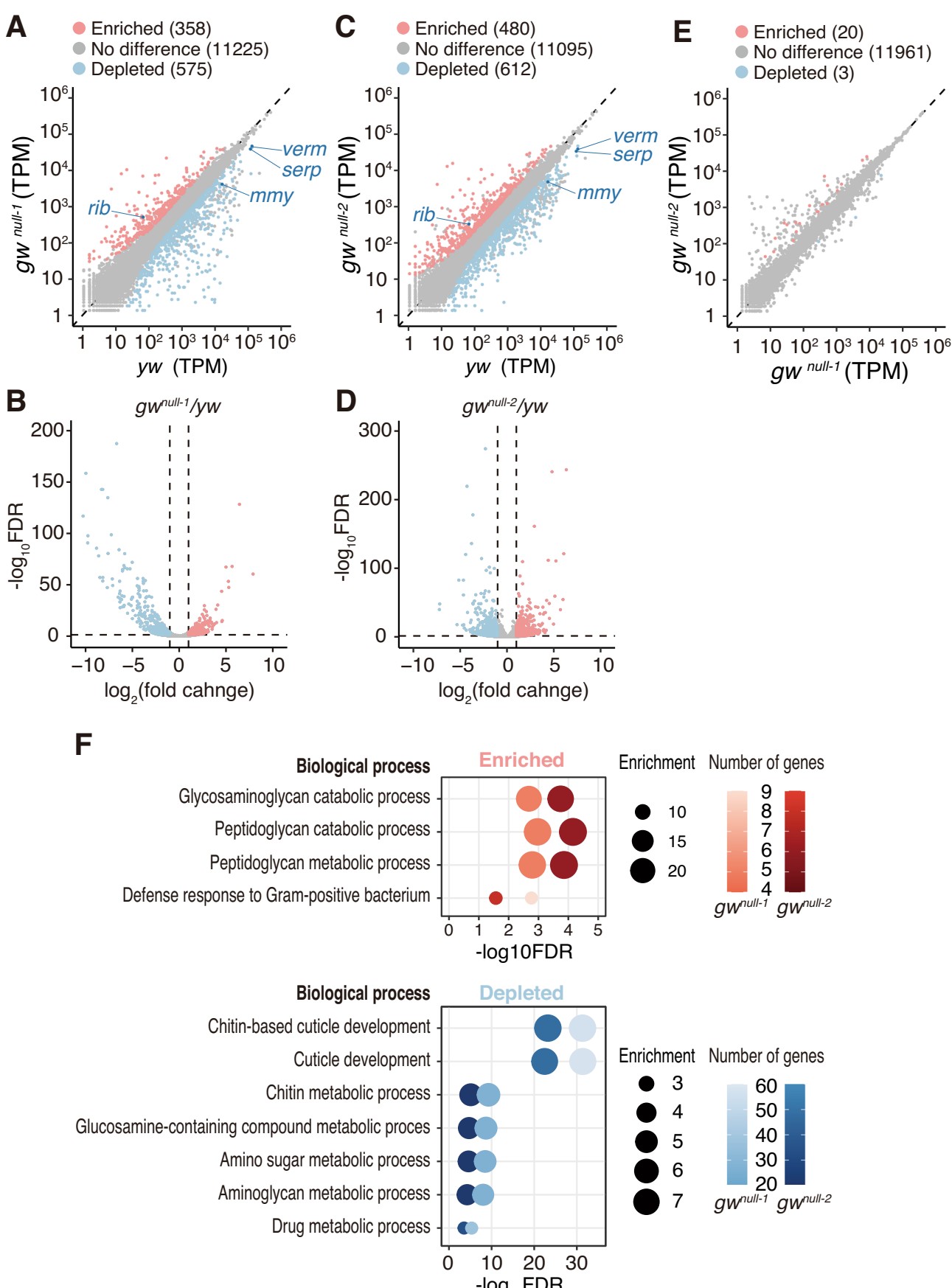

**Figure 6. GW182 depletion shows the downregulation of genes related to cuticle development.**

Gene expression comparison of $gw^{null-1}$ (**A, B**) or $gw^{null-2}$ (**C, D**) with $yw$ by RNA-seq. (**A, C**) Scatter plot (**B, D**) Volcano plot. The densities of RNA-seq reads on each protein-coding gene were calculated as transcripts per kilobase million (TPM). The average TPM of two biological replicates is shown. Compared to $yw$ RNA-seq, genes with log2FC >1 or log2FC < –1 (FC, fold change) with FDR < 0.05 (FDR, false discovery rate) in $gw^{null-1}$ or $gw^{null-2}$ RNA-seq were defined as enriched (red) or depleted (blue), respectively. The numbers of genes in each category (enriched, no difference, depleted) are shown in parentheses. (**E**) Gene expression comparison of $gw^{null-1}$ and $gw^{null-2}$ by RNA-seq and scatter plot. The densities of RNA-seq reads on each protein-coding gene were calculated as transcripts per kilobase million (TPM). The average TPM of two biological replicates is shown. Compared to $gw^{null-1}$ RNA-seq, genes with log2FC >1 or log2FC < –1 (FC, fold change) with FDR < 0.05 (FDR, false discovery rate) in $gw^{null-2}$ RNA-seq were defined as enriched (red) or depleted (blue), respectively. The numbers of genes in each category (enriched, no difference, depleted) are shown in parentheses. (**F**) Significantly enriched GO (Gene Ontology) categories among genes that are enriched or depleted in $gw^{null-1}$ and $gw^{null-2}$. The GO categories (FDR < 0.05) that are common between $gw^{null-1}$ and $gw^{null-2}$ are shown. The number of genes associated with a specific GO term in the RNA-seq data is shown by color gradient, and enrichment is shown by circle size. For details, see "Methods".

## Methods

### *Drosophila* strains

Fly culture and crosses were carried out at room temperature. *miR-8-GAL4* strain (#104917) and *w1118; PBac(681.P.FSVS-1)PMCA^CPTI001995* (#115256) were obtained from Kyoto *Drosophila* Genomics and Genetics Resource Center. *GMR-GAL4* strain (#9146), *ago1*-IR strain [*y1 v1; P(TRiP.HM04006)attP2* (#31700)] and *ago2*-IR [*y1 sc* v1 sev21; P{TRiP.HMS00108}attP2* (#34799)] were obtained from Bloomington Stock Center. *gw*-IR(1) strains [*w1118; P(GD10428)v45772* (#v45772)] and *gw*-IR(2) [*P(KK101472)VIE-260B* (#v103581)] were obtained from Vienna *Drosophila* Resource Center. *M(UAS-mir-8.S)ZH-86Fb* (#F001944) was obtained from FlyORF. *w1118;ln(4)CiD, CiDpanCiD/ Dp(2;4)eyD, AlpeyD:eyD* was a kind gift from Tetsuya Tabata (The University of Tokyo).

### Generation of reporter transgenic flies

The UAS-MCS-K10 3′ UTR region of pUASp (Rørth, 1998) was amplified by PCR and then inserted into the pBS vector using NEBuilder HiFi DNA Assembly Master Mix (NEB) to generate pBS-UASp. For the internal control, the mCherry sequence amplified by PCR from pmCherry (Clontech) was inserted into pBS-UASp (pBS-UASp-mCherry). Subsequently, the UASp-mCherry-K10 3′ UTR region of pBS-UASp-mCherry was amplified by PCR and inserted into pattB vector to generate pattB-UASp-mCherry. For the Sensor and Mutated GFP reporters, the corresponding miR-8 target sequence was inserted downstream of GFP in pAcGFP (Clontech) using the XhoI/NotI restriction enzyme site. Then, a DNA fragment containing GFP with the miRNA target sequence was inserted into pBS-UASp using the KpnI/SpeI restriction enzyme site (pBS-UASp-GFP-miRNA). Next, a fragment of UASp-GFP-miRNA target site was amplified from pBS-UASp-GFP-miRNA by PCR and inserted into pattB-UASp-mCherry to create pattB-UASp-GFP-miRNA-UASp-mCherry. To obtain reporter transgenic lines, pattB-UASp-GFP-miRNA-UASp-mcherry was injected into fly embryos in which phiC31 integrase is expressed in the germline (*y[1] w[1118]; PBac{y[+]-attP-3B} VK00002*, #9723 Bloomington Stock Center) (BestGene Inc.). F0 flies were crossed with *yw* and F1 progeny with red-eye color was balanced by *CyO*.

### Antibodies

Rabbit anti-GW182 antibody was generated by immunizing His-tagged recombinant GW182 (1−1061 aa) and the antisera were affinity-purified using a Protein A column (MBL). The mouse monoclonal anti-AGO1 antibody was a kind gift from Mikiko Siomi. Rabbit DYDDDDK Tag antibody was purchased from Cell Signaling (2368S).

### Immunohistology

Eye discs of third instar larvae were dissected in PBS, fixed in 2% formaldehyde, 0.1% TritonX-100 containing PBS at 25 °C for 20 min, and then washed three times in PBS containing 0.1% Tween 20 (PBST) at 25 °C for 5 min each. After blocking with PBST containing 10% donkey serum at 25 °C for one hour, eye discs were incubated with the primary antibodies, anti-GFP (1:100) (MBL) and anti-mRFP (1:100) (MBL) in PBST containing 10% donkey serum at 4 °C overnight, followed by washing in PBST three times. The eye discs were then incubated with the secondary antibodies conjugated with fluorescein, Alexa488 (1:200) (Thermofisher), or Alexa596 (1:200) (Thermofisher) for 2 h at 25 °C. The eye discs were washed in PBST and mounted in Vectashield antifade mounting medium (Vector Laboratories). Confocal microscopy was carried out on FV3000 confocal laser scanning system (Olympus) and the fluorescence intensity was assessed with Fiji software. *P* values were calculated by Student's *t* test (unpaired, two-sided) or one-way ANOVA with the post hoc Tukey HSD test with R or Excel. In Figs. 2C, 4B, EV2B and EV3B, *P* values were below the lower limit of the statistical calculation software, hence the exact value cannot be shown.

### qRT-PCR

Total RNA was purified from eye discs or first-instar larvae by FastGene RNA Basic Kit (Nippon Genetics). The cDNA was synthesized by using PrimeScript RT reagent Kit (TAKARA) using the random hexamer. qRT-PCR was performed by TB Green Premix Ex Taq II (TAKARA) using gene-specific primers and Thermal Cycler Dice Real Time System (TAKARA). The sequences of gene-specific primer set were shown as follows: *GFP*, 5′-CAA CAT CGA GGA TGG CAG CGT-3′ and 5′-CAT GTG ATC GCG CTT CTC GTT G-3′ ;*mCherry*, 5′-GAA CGG CCA CGA GTT CGA GA-3′ and 5′-CTT GGA GCC GTA CAT GAA CTG AGG-3′; *rp49*, 5′-CAC CAG TCG GAT CGA TAT GCT AAG-3′ and 5′-GTA ACC GAT GTT GGG CAT CAG A-3′; *gw182*, 5′-CTG CTT ACA CTG ATC TGG TCC AAG AGT TT-3′ and 5′-CGC TTC TTG GTG TAA TGC TGG GAT CAT CTT-3′, *ago1*, 5′-GCG AGG TTT GGT TCG GTT TC-3′ and 5′-CGT TGA TGT CGC GAA TGT CC-3′. For miRNA qPCR, total RNA was purified from first-instar

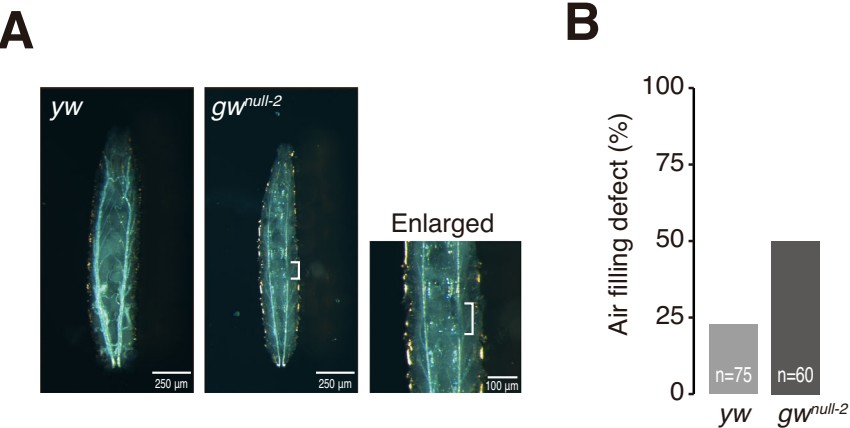

**Figure 7. Deletion of GW182 prevents larval trachea formation.**

(A) Ventral view of first-instar larva of *yw* or *gw^null-2* homozygote and a magnification of representative air-filling defect of *gw^null-2* homozygote. Defective air-filling region is indicated by a white bracket. (B) Percentage of air filling defect of *yw* or *gw^null-2* homozygote. Source data are available online for this figure.

larvae by miRNeasy kit (QIAGEN) according to the manufacturer's instructions. For Fig. 5C, the cDNA was synthesized using miScript II RT kit (QIAGEN). qRT-PCR was performed by QuantiTect SYBR Green PCR using miRNA-specific primers provided as miScript Universal primer (QIAGEN), Dm_miR-8_1 miScript Primer assay (QIAGEN), and Dm_miR-14_1 miScript Primer assay (QIAGEN) and Thermal Cycler Dice Real Time System (TAKARA). For Fig. EV5C, the cDNA was synthesized using miRCURY LNA RT Kit (QIAGEN) and qRT-PCR was performed by miRCURY LNA SYBR Green PCR Kit (QIAGEN) and Thermal Cycler Dice Real Time System (TAKARA) with dme-miR-8-3p miRCURY LNA miRNA PCR Assay (QIAGEN), dme-miR-14-3p miRCURY LNA miRNA PCR Assay (QIAGEN), and dme-U6-snRNA custom miRCURY LNA miRNA PCR Assay (QIAGEN). *P* values were calculated by Student's *t* test (unpaired, two-sided) or one-way ANOVA with the post hoc Tukey HSD test with R. In Figs. 2D and 4C, *P* values were below the lower limit of the statistical calculation software, hence the exact value cannot be shown.

## Generation of the *gw182*-null alleles

CRISPR/Cas9-mediated knockout of the *gw* locus was performed using the pU6-BbsI-chiRNA plasmid (Addgene # 45946) (Gratz et al, 2013). The target site in *gw182 gene* was selected by using CRISPR DESIGN site (http://crispr.mit.edu). The sense and antisense oligonucleotides corresponding to the selected target sequence were annealed and inserted into BbsI-digested pU6-BbsI-chiRNA plasmid. The *gw* chiRNA plasmid was injected into embryos of the *y^1 M(vas-Cas9)ZH-2A w^1118* strain (BestGene Inc.). Males or females carrying vas-Cas9 and a sgRNA transgene were crossed to *W^1118; ln(4)Ci^D, Ci^DpanCi^D/Dp(2;4)ey^D, Alpey^D:ey^D* flies. From each progeny, ten flies were isolated and checked for the deletion of *gw* by genome PCR. Eventually seven independent strains that have frameshift mutations in *gw182* were established. In this study, line 1 and 2, which had the largest deletion regions, were used and named *gw^null-1* and *gw^null-2*, respectively.

## Genome PCR

Genome DNA from a wing of the adult fly of each strain was extracted as follows. A wing was suspended in 5 μl of genome extraction buffer (25 mM NaOH, 0.2 mM EDTA) and incubated at 95 °C for 10 min, and then the suspension was neutralized by 5 μl of 150 mM HCl. 2 μl of the extracted genome solution was used as a template for Genome PCR by GoTaq (Promega). The sequences of *gw182* gene-specific primer set designed around the CRISPR target site were 5′-GAG TCC AAT TTG AGA AAC GGA GGT CA-3′ and 5′-CTG ATC GTT TGC GCT AAC TTC AT TAA TTC T-3′. The PCR products were analyzed by MultiNA (SHIMADZU). The PCR fragments that were shorter than that of the wild type were subcloned into pCR-TOPO (Invitrogen) and sequenced using the M13 forward and reverse primers.

## Determination of the lethal stage of *gw182*-null mutants

Eggs of the progeny of *gw^null-1/PBac(681.P.FSVS-1)PMCA^CPTI001995* or *gw^null-2/PBac(681.P.FSVS-1)PMCA^CPTI001995* were laid on grape juice plates with yeast pastes at 25 °C for 2 h. 26, 50 and 76 h after egg laying, each 88–96 larvae were collected and checked for the genotype by genome PCR. *PBac(681.P.FSVS-1)PMCA^CPTI001995*, expressing YFP trapped in the *PMCA* gene on the 4th chromosome (Lowe et al, 2014), was used as a marker.

## Generation of flies expressing truncated GW182

The genomic region 5185 bp upstream of the first ATG of *gw182* was cloned and joined with the full-length *gw182* CDS containing FLAG-tag in the C-terminal region, and the 3′ UTR. This construct was inserted into the pattB vector [pattB-(GW182-full)-flag]. For the truncated mutants of GW182, the CDS region was replaced by the corresponding GW182 truncated sequences [pattB-(GW182-605)-flag and pattB-(GW182-940)-flag]. In order to obtain transgenic lines, pattB-(GW182-full)-flag, pattB-(GW182-605)-flag or pattB-(GW182-940)-flag were injected into fly embryos in which phiC31 integrase is expressed in the germline (*y^1 M(vas-int.Dm)*

*ZH-2A w\*; M(3xP3-RFP.attP')ZH-51C*, #24482 Bloomington Stock Center) (BestGene Inc.). F0 flies were crossed with *yw* and F1 progeny with red-eye color was balanced by *CyO*.

### Rescue of *gw182*-null allele by GW182 truncated mutants

Females of $gw^{null1-2}$/ $ln(4)Ci^D$, $Ci^D panCi^D$ were crossed with males of *GW182-605 (II)*; $gw^{null-2}$/ $ln(4)Ci^D$, $Ci^D panCi^D$ or *GW182-940 (II)*; $gw^{null-2}$/ $ln(4)Ci^D$, $Ci^D panCi^D$ or *GW182-full (II)*; $gw^{null-2}$/ $ln(4)Ci^D$, $Ci^D panCi^D$ or +/+; $gw^{null-2}$/ $ln(4)Ci^D$, $Ci^D panCi^D$ (negative control). The F1 progeny with GW182 transgene was selected based on RFP expression, a marker of the attP site (Bischof et al, 2007), at each larval stage. Genome PCR was performed for selected larvae to discriminate homozygote and heterozygote of $gw^{null-2}$. The sequences of the *gw182* gene-specific primer set were 5′-GAG TCC AAT TTG AGA AAC GGA GGT CA-3′ and 5′-CAT TAA TTC TAA TAG ATA TTA GGC CTA-3′. The percentage of rescue was calculated by dividing the number of homozygotes of $gw^{null-2}$ with a truncated *gw182* transgene by the total number of larvae with the *gw182* transgene.

### Western blots and antibodies

For western blots of *gw182*-null larvae, YFP-negative first-instar larvae of the progeny of $gw^{null-1}$/$PBac(681.P.FSVS-1)PMCA^{CPTI001995}$ or $gw^{null1-2}$/$PBac(681.P.FSVS-1)PMCA^{CPTI001995}$ were collected 38 h after egg laying and were lysed in Passive lysis buffer (Promega). After centrifugation, proteins were loaded on an 8% polyacrylamide gel and transferred onto a PVDF membrane (Millipore). Chemiluminescence was induced by Luminata Forte Western HRP Substrate (Millipore), and images were acquired by a LAS-3000 imaging system (FujiFilm) or Amersham Imager 600 (GE Healthcare). Rabbit anti-GW182 (1:1000), mouse anti-Ago1 (1:1000), and goat anti-$\beta$-Actin (1:1000, Santa Cruz sc-1615) were used for blotting. *yw* first-instar larvae were used as a positive control of GW182 expression.

### RNA-seq

First-instar larvae were collected and extracted total RNA by RNA basic kit (FastGene) according to the manufacturer protocol. For Figs. 5 and 6, total RNA libraries were prepared by using SMART-Seq v4 Ultra® Low Input RNA Kit for Sequencing (Clontech) according to the manufacturer's instruction and analyzed by Illumina HiSeq 2500 system (Illumina). For Fig. EV5, total RNA libraries were prepared using the Strand-Specific mRNA Library Preparation method by BGI. mRNA was first purified using oligo dT-beads, fragmented, and then used to synthesize cDNA. The resulting double-stranded cDNA fragments underwent end-repair, followed by the addition of a single 'A' nucleotide to the 3′ ends of the blunt fragments. Next, adaptors were ligated to the cDNAs, which were subsequently amplified by PCR. Finally, the cDNA library was analyzed using the DNBseq system by MGI Tech.

RNA sequence reads were analyzed on the Galaxy web platform (usegalaxy.eu) (Afgan et al, 2018) or by manual commands. After trimming of the adaptor sequence, quality was controlled by FastQC 0.11.2. Reads were mapped to *Drosophila melanogaster*

BDGP6.28 genome using STAR (version 2.7.0a) (Dobin et al, 2013). For the differential expression analysis at the transcripts level, mapped reads were counted using RSEM (Li and Dewey, 2011). For the differential expression analysis at the gene level, mapped reads were counted using *featureCounts* (Liao et al, 2014) by summing the total number of reads overlapping at least ten nucleotides with the gene annotation, including only uniquely mapped reads in the genome. Differential gene expression and normalized counts as TPM (Transcripts Per Kilobase Million) were calculated using *DESeq2* (Love et al, 2014). Genes having >0 TPM were kept for further analysis. Processed data were transferred to R, then cumulative fractions of fold change values were drawn with the ggplot2 package. The FDR was calculated using the Benjamini and Hochberg method. Target mRNAs of each miRNA were predicted by using TargetScan (Fly, Release 7.2) (Agarwal et al, 2018). Gene ontology analysis was performed using stringent criteria ($\log_2$FC [fold change] > 1 or < –1; FDR [false discovery rate] <0.05) by GOrilla (Eden et al, 2009). The Protein–protein networks in enriched or depleted biological processes were analyzed by STRING (Szklarczyk et al, 2019). For Fig. 6F, *N*: the total number of genes. *B*: the total number of genes associated with a specific GO term. *n*: the total number of genes in the RNA-seq data. *b*: the number of genes associated with a specific GO term in the RNA-seq data, shown by color gradient. Enrichment = (b/n)/(B/N), shown by circle size.

### Assessment of reporter expression in first-instar larvae

First-instar larvae were collected after 40–42 h of hatching and the fluorescence of GFP and mCherry under fluorescence microscopy were observed (Zeiss). Fluorescence intensity was quantified by Fiji software. For each larva, the average fluorescence intensity of GFP or mCherry in the body part of interest was measured and the average of auto-fluorescence of *yw* was subtracted as the background.

### Trachea observation

Second-instar larvae after 50 h after egg laying were collected and fixed with 4% Paraformaldehyde/PBST. After washing with PBST, the trachea was observed under microscopy.

### Accession numbers

The RNA-seq data obtained in this study (GSE192811 and GSE267756) were deposited to the National Center for Biotechnology Information (NCBI).

## Data availability

The datasets and computer code produced in this study are available in the following databases: RNA-Seq data: Gene Expression Omnibus GSE192811. RNA-Seq data: Gene Expression Omnibus GSE267756.

The source data of this paper are collected in the following database record: biostudies:S-SCDT-10_1038-S44318-024-00249-4.

# Peer review information

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

## Acknowledgements

The authors thank all the members of the Tomari laboratories for constructive discussion, technical help, and critical reading of the manuscripts. RNA sequencing was supported by the IQB central laboratory, The University of Tokyo. This work was supported by JSPS KAKENHI (18H05271 and 21H05278 to YT, 21K06026 to EMS), RIKEN Pioneering Project, and Aging Project to SI.

## Author contributions

**Eriko Matsuura-Suzuki**: Conceptualization; Formal analysis; Funding acquisition; Investigation; Visualization; Methodology; Writing—original draft; Writing—review and editing. **Kaori Kiyokawa**: Investigation. **Shintaro Iwasaki**: Resources; Supervision; Funding acquisition. **Yukihide Tomari**: Conceptualization; Resources; Supervision; Funding acquisition; Methodology; Project administration; Writing—review and editing.

Source data underlying figure panels in this paper may have individual authorship assigned. Where available, figure panel/source data authorship is listed in the following database record: biostudies:S-SCDT-10_1038-S44318-024-00249-4.

## Disclosure and competing interests statement

The authors declare no competing interests. Yukihide Tomari is a member of the Advisory Editorial Board of *The EMBO Journal*. This has no bearing on the editorial consideration of this article for publication.

# Expanded View Figures

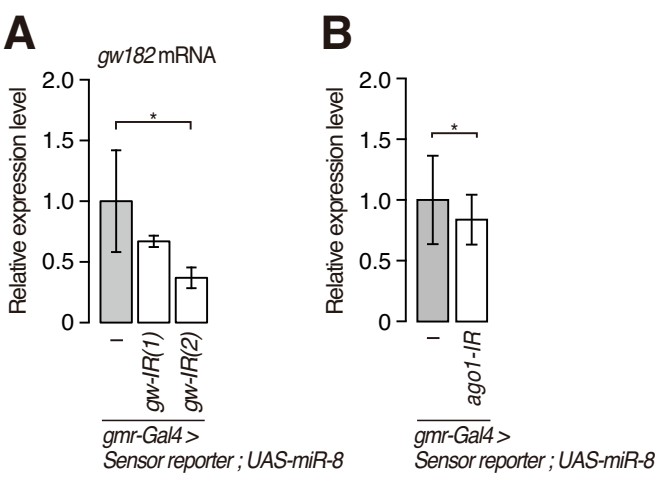

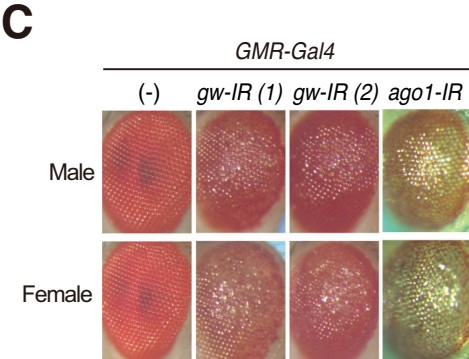

**Figure EV1. Quantification of *gw182* mRNA expression level in *gw182* knockdown flies.**

(A) Quantification of *gw182* mRNA expression. The *gw182* mRNA levels of two independent *gw182*-IR-expressing strains, *gw-IR(1)* and *gw-IR(2)*, at the third instar larval stage were measured by qRT-PCR. *rp49* was used as a reference. The relative expression levels against the control are shown as mean ± SD. Error bars represent standard derivation from three independent experiments. *P* values were calculated by one-way ANOVA with the post hoc Tukey HSD test. *$P = 0.047$. (B) Quantification of *ago1* mRNA expression. The *ago1* mRNA levels of *ago1*-IR-expressing strains at the third instar larval stage were measured by qRT-PCR. rp49 was used as a reference. The relative expression levels against the control are shown as mean ± SD. Error bars represent standard derivation from three independent experiments. *P* values were calculated by Student's *t* test (unpaired, two-sided). *$P = 0.538$. (C) Rough eye phenotype by *gw182* or *ago1* knockdown. The eye phenotypes of two independent *gw182*-IR-expressing strains, *gw-IR(1)* and *gw-IR(2)*, and *ago1*-IR-expressing strain at the adult stage were observed. *gmr-G4* strain was used as a control. Genotypes: *GMR-GAL4/ +* , *GMR-GAL4,GW182-IR(1)/ +* , *GMR-GAL4,GW182-IR(2)/ +* , *GMR-GAL4,ago1-IR/+*.

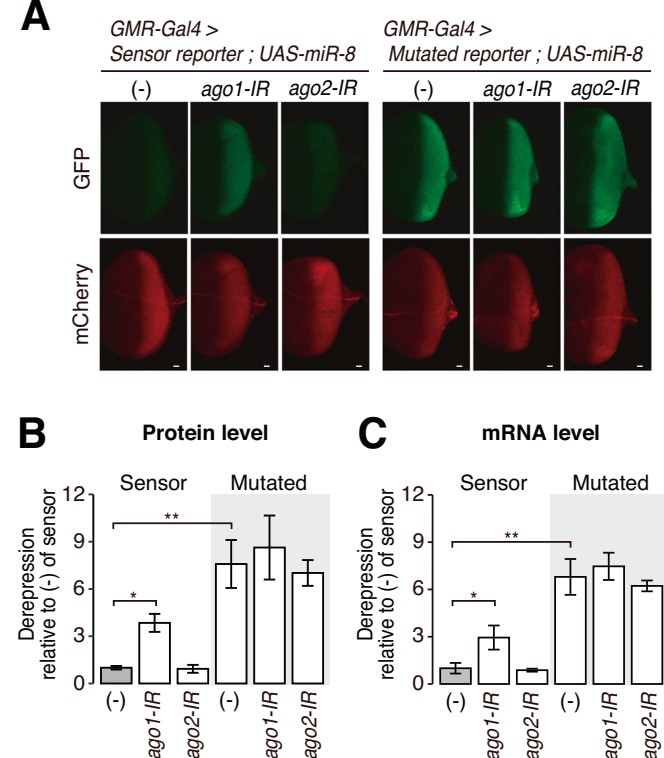

**Figure EV2.  Ago2 is not essential for miR-8-mediated silencing of the reporter in eye discs.**

(A) Fluorescence microscopy images for the Sensor reporter (A) and the Mutated reporter (B) upon miR-8 overexpression. *ago1* or *ago2* was knocked down by corresponding IR in the third instar larval eye disc. Scale bar: 20 µm. (B) Quantified derepression at the protein level in Fig. EV1A. The GFP fluorescence level was normalized to the mCherry fluorescence level, and the derepression levels are shown relative to the negative control without IR expression (–) of the Sensor reporter or the Mutated reporter as mean ± SD. Error bars represent standard derivation from 10 independent eye discs for each sample. *P* values were calculated by one-way ANOVA with the post hoc Tukey HSD test. *$P = 0.78e-5$; **$P < 1.00e-7$. (C) Quantified derepression at the mRNA level in the Sensor and Mutated reporters. GFP mRNA level was normalized to mCherry mRNA level, and the derepression levels are relative to the negative control (–) of the Sensor reporter, shown as mean ± SD. Error bars represent standard derivation from 3 independent experiments. *P* values were calculated by one-way ANOVA with the post hoc Tukey HSD test. *$P = 4.31e-2$; **$P = 3.20e-6$. Genotypes: *GMR-GAL4/UAS-Sensor reporter; UAS-miR-8/+* (n = 10), *GMR-GAL4,ago1-IR/UAS-Sensor reporter; UAS-miR-8/+* (n = 10), *GMR-GAL4,ago1-IR/UAS-Sensor reporter; UAS-miR-8/ago2-IR* (n = 10), *GMR-GAL4/UAS-Mutated reporter; UAS-miR-8/+* (n = 10), *GMR-GAL4,ago1-IR/UAS-Mutated reporter; UAS-miR-8/+* (n = 10), *GMR-GAL4,ago1-IR/UAS-Mutated reporter; UAS-miR-8/ago2-IR* (n = 10). The numbers (n) are for Fig. EV2D.

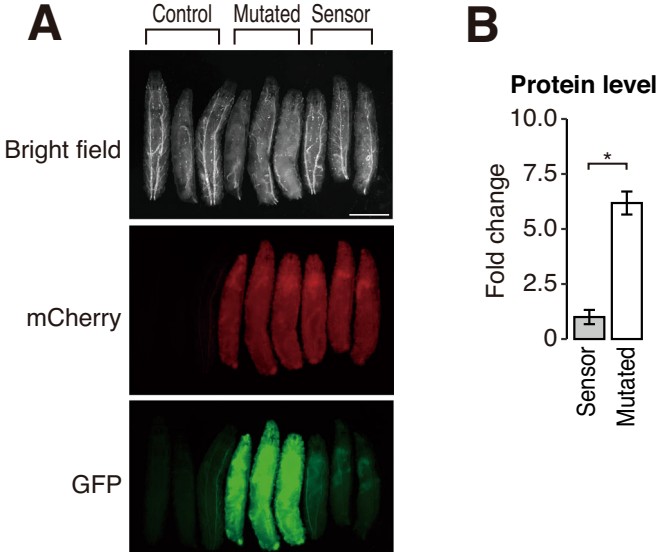

**Figure EV3. The Sensor reporter monitors endogenous miR-8 silencing activity at the larval stage.**

(A) Fluorescence microscopy of the Sensor reporter and the Mutated reporter at first-instar larval stage. Bright field, GFP (green), or mCherry (red) expressions are compared between *yw*, the Sensor reporter strain, or the Mutated reporter strain. *yw* was used as a negative control. Scale bar: 250 μm. (B) Quantification of the GFP protein level for Fig. EV3A. The fluorescence GFP levels were normalized to the fluorescence mCherry levels, shown as mean ± SD. Error bars represent standard derivation from three independent larvae for each sample and *P* values were calculated by Student's *t* test (unpaired, two-sided). *$P < 2.2e\text{-}16$. Genotypes: *yw, miR-8-GAL4/UAS-Sensor reporter* ($n = 42$) and *miR-8-GAL4/UAS-Mutated reporter* ($n = 47$). The numbers ($n$) are for Fig. EV3B.

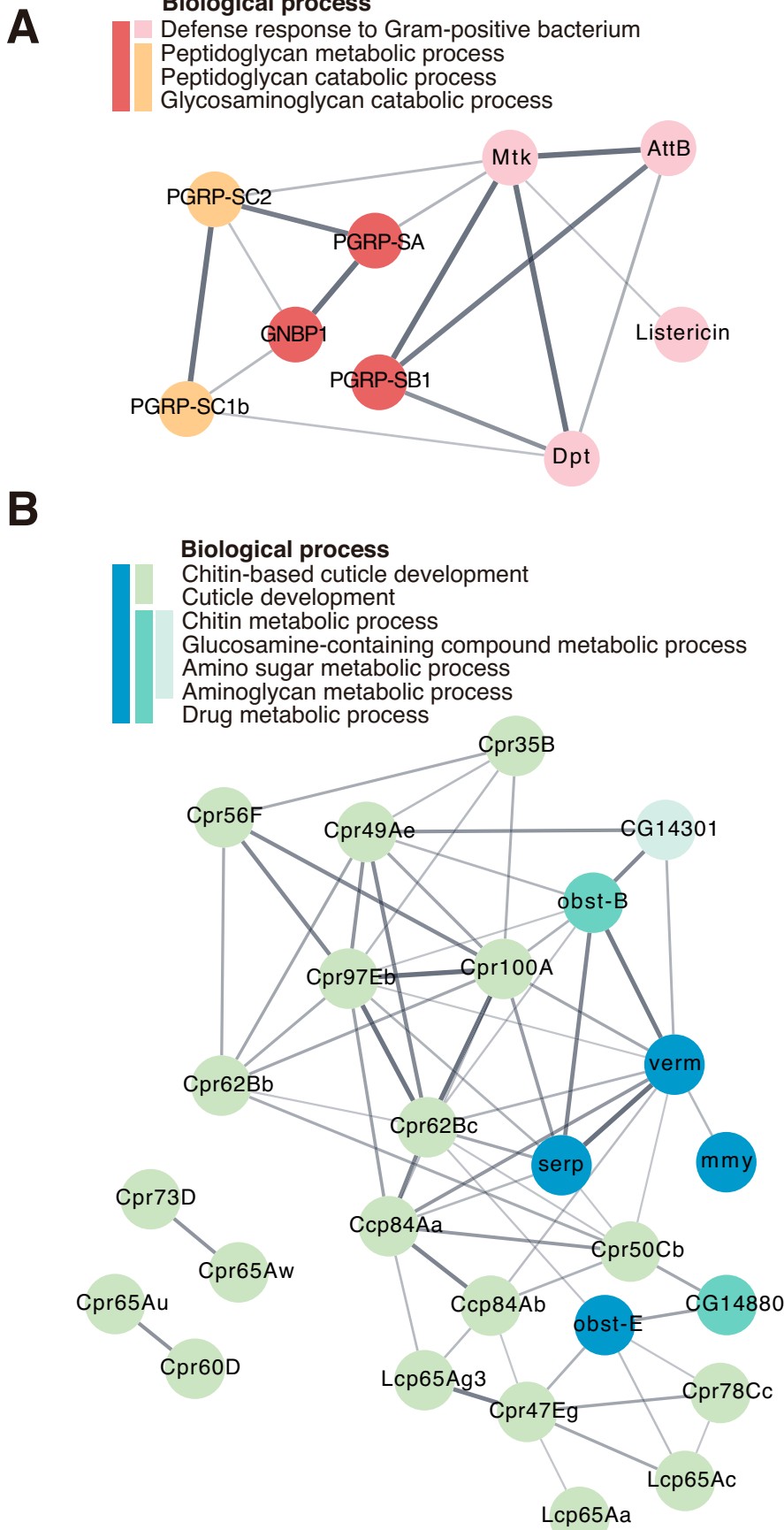

**A**

**Biological process**
- Defense response to Gram-positive bacterium
- Peptidoglycan metabolic process
- Peptidoglycan catabolic process
- Glycosaminoglycan catabolic process

**B**

**Biological process**
- Chitin-based cuticle development
- Cuticle development
- Chitin metabolic process
- Glucosamine-containing compound metabolic process
- Amino sugar metabolic process
- Aminoglycan metabolic process
- Drug metabolic process

◄  **Figure EV4.  Networks of *gw182*-regulated genes and related biological processes.**

(A) A protein–protein association network of enriched genes in *gw182*-null mutant and their associated functional classes. Genes classified to defense to gram-positive bacterium, pink; genes classified to peptidoglycan metabolic process, peptidoglycan catabolic process and glycosaminoglycan catabolic process, orange; genes classified to the all above GO terms, red. (B) A protein–protein association network of depleted genes in *gw182*-null mutant and their associated functional classes. Genes classified to chitin-based cuticle development and cuticle development, lime green; Genes classified to chitin metabolic process, glucosamine-containing compound metabolic process, amino sugar metabolic process, and aminoglycan metabolic process, light blue; genes classified to GO terms represented by light blue and drug metabolic process, aqua green; genes classified to all above GO terms, aqua blue.

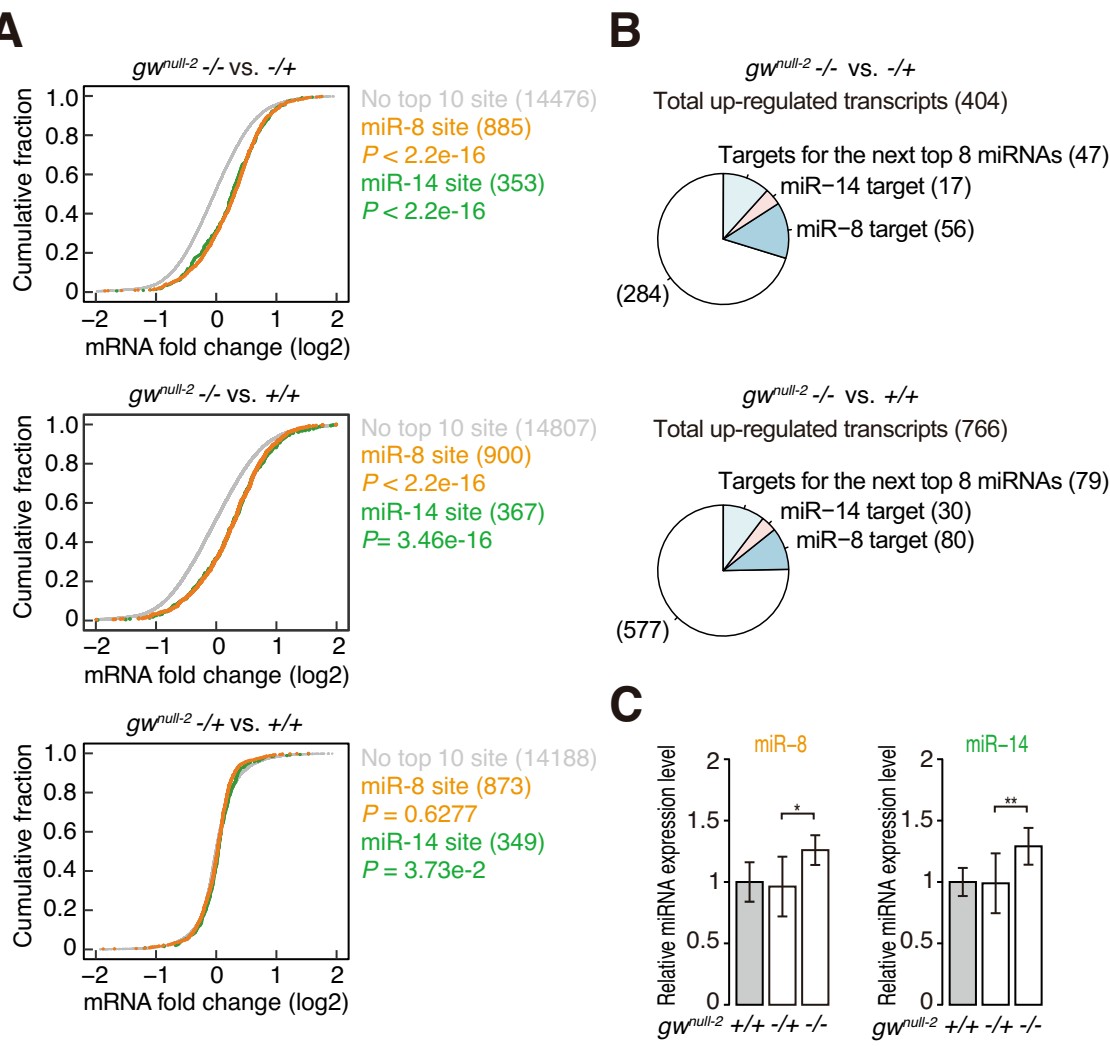

**Figure EV5.   Depletion of GW182 impaired miRNA-mediated silencing for endogenous miRNA targets at the first-instar larval stage, related to Fig. 5.**

(**A**) Upregulation of miRNA targets by knockdown of *gw182*. First-instar larvae of *gw^{null-2}* homozygote, *gw^{null-2}* heterozygote or wild type were collected among siblings, and their transcriptome was analyzed by RNA-seq. The fold change values, *gw^{null-2}* homozygote/*gw^{null-2}* heterozygote, *gw^{null-2}* homozygote/wild type, or *gw^{null-2}* heterozygote/wild type were calculated. The cumulative fractions of the average fold change values of two independent experiments are shown for the following categories: mRNAs without a target site of the top 10 most abundant miRNAs (gray) and mRNAs with predicted target sites for miR-8 (orange) or miR-14 (green), together with their *P* values of the Mann–Whitney *U* test. *P* values are shown in the figure. (**B**) Pie chart showing the percentage of predicted miRNA targets within the transcripts upregulated in *gw^{null-2}* homozygote compared to *gw^{null-2}* heterozygote or wild type among siblings. The numbers of transcripts in each category are shown in parentheses. (**C**) Quantification of microRNA expression. The relative expression levels of miR-8 and miR-14 were measured by qRT-PCR from first-instar larvae of *gw^{null-2}* homozygote, *gw^{null-2}* heterozygote, or wild type among siblings. *U6 snoRNA* was used as a reference. The data were normalized by the miRNA expression level of wild type, shown as mean ± SD. Error bars represent standard derivation from three independent experiments. *P* values were calculated by Student's *t* test (unpaired, two-sided). *$P = 0.106$; **$P = 0.092$.

