## [Peer Review File · The EMBO Journal]

miRNA-mediated gene silencing in *Drosophila* larval development involves GW182-dependent and independent mechanisms

Eriko Matsuura-Suzuki, Kaori Kiyokawa, Shintaro Iwasaki, and Yukihide Tomari

Corresponding author(s): Yukihide Tomari (tomari@iqb.u-tokyo.ac.jp)

Review Timeline:

Submission Date:	10th Mar 22
Editorial Decision:	4th Apr 22
Revision Received:	17th May 24
Editorial Decision:	25th Jun 24
Revision Received:	11th Aug 24
Accepted:	11th Sep 24

Editors: Stefanie Boehm and Cornelius Schneider

Transaction Report:

Prof. Yukihide Tomari
The University of Tokyo
Institute for Quantitative Biosciences
1-1-1 Yayoi
Life Sciences Research Building B-409
Bunkyo-ku, Tokyo 113-0032
Japan

4th Apr 2022

Re: EMBOJ-2022-111126
The requirement of GW182 in miRNA-mediated gene silencing in Drosophila larval development

Dear Prof. Tomari,
dear Yuki,

Thank you for submitting your manuscript on the role of GW182 in Drosophila larval development to The EMBO Journal. We have now received three referee reports on your study, which are included below for your information. In light of the referees' comments, we would like to invite you to prepare and submit a revised manuscript.

As you will see, the reviewers are overall positive and appreciate the findings and their interest to the field. However, they also raise a number of points that should be addressed before the study is considered further for publication. In particular, both referee #1 and referee #3 raise concerns with respect to the results and conclusions based on the miR-8 and miR-14 reporter experiments, and this should be discussed further and additional experimental data added where appropriate (ref#1- point 3, 4, 5; ref#3- point 3). In addition, GW182-dependent and -independent miRNA function should be further defined to address referee #1's (point 1, 2) and referee #2's (point 2, 3) concerns, possibly by comparing Ago2 and Ago1 as referee #1 suggests. Please also carefully consider all other referee comments and revise the manuscript and figures as needed, as well as providing a detailed response to each comment.

Please note that it is our policy to allow only a single round of major revision. Acceptance depends on a positive outcome of a second round of review and therefore on the completeness of your responses included in the next, final version of the manuscript. I encourage you to review the referees' comments and contact me to discuss any questions or a preliminary revision plan in case there are any uncertainties regarding specific points or the revision in general.

Thank you for the opportunity to consider your work for publication. I look forward to receiving your revised manuscript!

Kind regards,

Stefanie

Stefanie Boehm
Editor
The EMBO Journal

***IMPORTANT NOTE: we now perform an initial quality control of all revised manuscripts before re-review. Your manuscript will FAIL this control and the handling will be DELAYED if the following APPLIES:

- 1) A data availability section is missing.
- 2) Your manuscript contains error bars based on $n=2$. Please use scatter blots showing the individual datapoints in these cases. The use of statistical tests needs to be justified.

When submitting your revised manuscript, please carefully review the instructions that follow below. Failure to include requested items will delay the evaluation of your revision.***

-

When submitting your revised manuscript, please carefully review the instructions below and include the following items:

- 1) A .docx formatted version of the manuscript text (including legends for main figures, EV figures and tables). Please make sure that the changes are highlighted to be clearly visible.
- 2) Individual production quality figure files as .eps, .tif, .jpg (one file per figure).
- 3) A .docx formatted letter INCLUDING the reviewers' reports and your detailed point-by-point response to their comments. As part of the EMBO Press transparent editorial process, the point-by-point response is part of the Review Process File (RPF), which will be published alongside your paper.
- 4) A complete author checklist, which you can download from our author guidelines (<https://wol-prod-cdn.literatumonline.com/pb-assets/embo-site/Author%20Checklist%20-%20EMBO%20J-1561436015657.xlsx>). Please insert information in the checklist that is also reflected in the manuscript. The completed author checklist will also be part of the RPF.
- 5) All corresponding authors are required to supply an ORCID ID for their name upon submission of a revised manuscript.
- 6) Before submitting your revision, primary datasets produced in this study need to be deposited in an appropriate public database. Please remember to provide a reviewer password if the datasets are not yet public.

Please see the instructions below on how to format the Data Availability section (see <https://www.embopress.org/page/journal/14602075/authorguide#datadeposition>). Please note that the Data Availability Section is restricted to new primary data that are part of this study.

Data availability

[data type]: [name of the resource] [accession number/identifier/doi] ([URL or identifiers.org/DATABASE:ACCESSION])

7) Our journal encourages inclusion of *data citations in the reference list* to directly cite datasets that were re-used and obtained from public databases. Data citations in the article text are distinct from normal bibliographical citations and should directly link to the database records from which the data can be accessed. In the main text, data citations are formatted as follows: "Data ref: Smith et al, 2001" or "Data ref: NCBI Sequence Read Archive PRJNA342805, 2017". In the Reference list, data citations must be labeled with "[DATASET]". A data reference must provide the database name, accession number/identifiers and a resolvable link to the landing page from which the data can be accessed at the end of the reference. Further instructions are available at < <https://www.embopress.org/page/journal/14602075/authorguide#referencesformat> >.

8) We would also encourage you to include the source data for figure panels that show essential data. Numerical data can be provided as individual .xls or .csv files (including a tab describing the data). For 'blots' or microscopy, uncropped images should be submitted (using a zip archive or a single pdf per main figure if multiple images need to be supplied for one panel).

Please note that source data should be uploaded in the following format:

- a) For main figures: please upload the source data as one zip file per figure and label it as source_data_figX.
- b) For EV figures please compile individual (zip) files for each figure, then combine these and upload one zip file for all figures. Please label this as source_data_EVfigures.
- c) For Appendix figures please also make one file per figure, combine all Appendix files into one final zip file for all figures and label it as source_data_Appendix.

Additional information on source data and instruction on how to label the files are available at < <https://www.embopress.org/page/journal/14602075/authorguide#sourcedata> >.

9) We replaced Supplementary Information with Expanded View (EV) Figures and Tables that are collapsible/expandable online (see examples in <http://msb.embopress.org/content/11/6/812>). A maximum of 5 EV Figures can be typeset. EV Figures should be cited as 'Figure EV1, Figure EV2' etc.. in the text and their respective legends should be included in the main text after the legends of regular figures.

- For the figures that you do NOT wish to display as Expanded View figures, they should be bundled together with their legends in a single PDF file called *Appendix*, which should start with a short Table of Content. Appendix figures should be referred to in the main text as: "Appendix Figure S1, Appendix Figure S2" etc. See detailed instructions regarding expanded view here: < <https://www.embopress.org/page/journal/14602075/authorguide#expandedview> >.

https://wol-prod-cdn.literatumonline.com/pb-assets/embo-site/EMBOPress_Figure_Guidelines_061115-1561436025777.pdf

Further information is available in our Guide For Authors:

We realize that it is difficult to revise to a specific deadline. In the interest of protecting the conceptual advance provided by the work, we recommend a revision within 3 months (3rd Jul 2022). Please discuss the revision progress ahead of this time with the editor if you require more time to complete the revisions. Use the link below to submit your revision:

Link Not Available

Referee #1:

Matsuura-Suzuki and Tomari report here the characterization of a newly made "clean" null allele of the gw182 gene in *Drosophila*. In contrast to the previously available gw182-null allele, which contained additional mutations, they show that gw182-null flies can survive until the second instar larval stage, and, in the absence of GW182, a miRNA-specific reporter can be repressed. They also provide some insights into the biological role of GW182 at the later stage of development by performing comparative RNA-seq and monitoring physiological defects in larvae.

Overall, this work is quite interesting, as it confirms the presence and the conservation of GW182-dependent and -independent miRNA-mediated silencing mechanisms in animals, as previously reported in the roundworm. It also provides to the community a true null allele of gw182 in flies that will be indeed useful for future analyses. With the following additional experiments/clarifications, this study can be a good candidate for publication:

-The authors suggest the presence of GW182-dependent and -independent miRNA-mediated silencing in flies but the data obtained by RNAi knockdown and with their new gw182-null alleles only show the effect with their new GW182-independent miRNA reporter. Therefore, it will be important for this study to show that a GW182-dependent reporter is sensitive to, at least, the loss of gw182 using this new CRISPR/Cas9 KO alleles.

-As nicely shown by this group before, while both Argonautes can bind miRNAs and repress mRNA targets in flies, GW182 interacts only with Ago1 but not Ago2. Thus, it will be relevant here to test and compare how Ago2 vs Ago1 contributes to the repression of their new GW182-independent miR-8 reporter. That will help gather some mechanistic insights about how the G182-independent silencing mechanism works in vivo.

-The authors should discuss the potential mechanism involved in the GW182-independent repression seen with the miR-8 reporter, which appears different from those observed with miR-8 and miR-14 predicted targets. In the absence of gw182, the mRNA level of the miR-8 reporter is not affected (Figures 2D and 4C), while the level of putative miR-8 and miR-14 targets in larvae at the first instar stage is significantly increased in gw182-null animals (Figure 5).

-It is also unclear why the miR-8 reporter mRNA level is increased in the absence of Ago1. The current model suggests that GW182 recruits CCR4-Not complex to induce degradation of microRNA targets. So why the silencing of this GW182-independent reporter will still be affected at the mRNA level in Ago1-null animals?

-Is there a reason for creating new nucleotides pairing after the seed region in the mutated miR-8 sensor? I am concerned that this new base-pairing can make this reporter now sensitive to Ago2 slicing activity, which is likely not the case for the non-mutated sensor.

-Please proof edit the figure legends. Some typos and mistakes can be found.

Referee #2:

Matsuura-Suzuki and Tomari present data examining the requirement of GW182 for miRNA-mediated gene regulation and larval development in *Drosophila*. The manuscript is clear, thoughtful, and very well crafted. Major findings and contributions include:

- creation of a well-controlled in vivo reporter for miR-mediated repression
- evidence that AGO1-RISC can effectively silence miRNA targets in fly eye discs even when GW182 levels are reduced
- generation of two GW182-null *Drosophila* alleles
- GW182-null flies can survive embryogenesis and first instar larval development
- GW182 is essential for development at an early second instar stage
- miR-8 can silence the in vivo reporter in GW182-null first instar larvae
- derepression of miR-8 and miR-14 targets in GW182-null larvae
- GW182-null larvae suffer misregulation of chitin-related genes and trachea system defects

The study is exciting and timely as several groups have reported GW182-independent silencing in vitro in recent years and yet, apart from the single study in *C. elegans* (PMID: 27935964), in vivo evidence is lacking. The study has the potential for a high impact on the miRNA field as it may direct researchers to consider mechanisms and functions for miRNAs outside of GW182, and may help mitigate against the tenuous, but sometimes accepted, assumption that all miRNAs function through GW182 family proteins (i.e. PMID: 34463618).

Because GW182 null flies do not develop to maturity, it is not possible to generate a stable homozygous knock-out line. Therefore, the major conclusions of the study hinge on the validity of the notion that miRNA function in the GW182-null larvae is not dependent on small, but functional, materially-deposited GW182 or gw182 mRNA reserves. Major concerns/suggestions:

- 1) Page 6: "Indeed, the expression of GW182 protein was at an undetectable level in the first instar larvae of gw null-1 and gw null-2 (Figure 3E)". Staring at Fig. 3E with my eyes slightly defocused, I believe I can detect a faint band around the size of GW182 in the null-1 lane and especially the null-2 lane. What does an over-exposed image of this blot look like? Can the authors quantitate band intensities to associate a numerical value with the limit of detection for GW182 protein levels?
- 2) An obvious orthogonal experiment would involve the creation of flies with mutations in the tryptophan-binding pockets in Ago1, as in PMID: 27935964. In this case, the Ago1 K785S + R813S double mutant would be expected to be impaired in binding GW motif I, which appears to be the primary Ago1-binding site in GW182 (PMID: 19383769, Fig. 1). Alternatively, a motif I GW182 mutant (i.e. W11A + W22A) is expected to be impaired in Ago1-binding. Have the authors considered experiments along these lines to support their conclusions regarding GW182-independent miRNA function?
- 3) Page 6: "These results suggest that AGO1-RISC can silence miRNA targets independently of GW182 in the fly eye disc." This suggestion seems rather strong considering that the knock-down of gw182 mRNA by the inverted repeats is only ~2-fold (Fig. S1). How does this compare to the knock-down ago1 in the ago1-IR line? Is ago1 knock-down similar?

Minor comments/suggestions:

- 4) Page 3: "The N-terminal glycine/tryptophan (GW) repeats of GW182 directly binds the two hydrophobic pockets (tryptophane-binding pockets) in the PIWI domain of AGO proteins (Elkayam et al., 2017; Schirle et al., 2014)" A third AGO protein tryptophan-binding pocket that directly binds GW repeats in TNRC6B was reported in PMID: 29576456.
- 5) Page 6: "It has been demonstrated that maternally deposited GW182 is depleted by 60-70 min after embryo laying (Schneider et al., 2006)." I am not sure this statement accurately reflects the data in Fig. 4 of Schneider et al. It looks to me that 60-70 min after egg deposition GW182 protein levels are about 50% of the level 0-10 min after egg deposition (Schneider et al., Fig. 4B, lower panel). GW182 levels then rise, which the authors attribute to zygotic transcription, making the fate of materially deposited GW182 after the 60-70 min window unknown. Are these the data to which the authors are referring, or am I missing something here?
- 6) It might be helpful to display Fig. S3 alongside data in Fig. 6A,C (this is only a suggestion).

Referee #3:

In the manuscript "The requirement of GW182 in miRNA-mediated gene silencing in *Drosophila* larval development," Suzuki and Tomari re-evaluate the previously addressed role of GW182 in miRNA-mediated gene regulation during *Drosophila* development. Based on two newly generated CRISPR-Cas9 genome editing-derived gw182 null alleles, they report that flies depleted of GW182 survive until an early second instar larval stage. This is in contrast to previous findings employing a truncation mutation in the gw182 locus, which indicated a requirement to enter gastrulation. Furthermore, GW182 appears to be dispensable for miRNA-mediated gene repression, since gw182 null flies were capable of repressing a heterologous reporter construct. At the same time, targets of abundant miRNAs in fly larvae are stabilized in the absence of GW182 confirming its contribution to reach a full extent or robustness of miRNA-mediated mRNA repression. Finally, characterization of the steady-state transcriptome in first instar larvae suggests a downregulation of chitin-related genes and air filling defects of trachea upon depletion of GW182, demonstrating the importance of GW182 in trachea formation. Overall, the study addresses an important

genetic conundrum in the genetic dependencies of GW182 in various model systems and implies a specific (possibly miRNA-independent) function in larval development. I therefore recommend publication of the manuscript if the following comments/concerns can be adequately addressed.

Comments:

- RNAseq analyses compare transcript abundances between gw182 null mutant flies and yw control flies. Hence, it is unclear to what extent the observed changes (i.e. upregulation of immune system-related transcripts and downregulation of chitin-related genes) are caused by loss-of-GW182 or different genetic background and/or stock-effects. I would strongly recommend to include a heterozygous sibling control in those experiments to distinguish these possibilities.
- It would be helpful to comment on the relationship between upregulated mRNAs and the presence/absence of target sites for highly abundant miRNAs to get an impression on the relationship of transcriptome changes and miRNA-related GW182-biology.
- It is interesting that miR-8 and miR-14 are significantly upregulated in gw182 null mutants. While those changes seem statistically significant, the authors refrain from including a possible explanation. Like mentioned above, this may be caused by genetic background, which can be addressed by involving comparisons to heterozygous siblings. Alternative explanations are certainly possible.
- Precise p-Values of changes in protein or RNA expression (etc.) that are deemed significant and non-significant should be reported to allow for unbiased interpretation.
- Figure 3d: n-values should be indicated to understand the population size that is underlying those data.
- Consider correcting in the first paragraph of discussion: "We found that, even in the absence of zygotic expression GW182, flies can complete the embryogenesis and develop into second instar larvae."

We thank all the Referees for their positive evaluation of our manuscript. First of all, we would like to begin by apologizing for the significant delay in revising our manuscript, which was in part caused by the impact of the pandemic and also the career transition of the first author from the academia to the industry. Despite these challenges, your constructive suggestions have been instrumental in guiding us to obtain new data that further strengthen our manuscript. We are deeply grateful for your insightful feedback.

Point-by-point Responses to the Referees' Comments

Referee #1:

Matsuura-Suzuki and Tomari report here the characterization of a newly made "clean" null allele of the *gw182* gene in *Drosophila*. In contrast to the previously available *gw182*-null allele, which contained additional mutations, they show that *gw182*-null flies can survive until the second instar larval stage, and, in the absence of GW182, a miRNA-specific reporter can be repressed. They also provide some insights into the biological role of GW182 at the later stage of development by performing comparative RNA-seq and monitoring physiological defects in larvae.

Overall, this work is quite interesting, as it confirms the presence and the conservation of GW182-dependent and -independent miRNA-mediated silencing mechanisms in animals, as previously reported in the roundworm. It also provides to the community a true null allele of *gw182* in flies that will be indeed useful for future analyses. With the following additional experiments/clarifications, this study can be a good candidate for publication:

-The authors suggest the presence of GW182-dependent and -independent miRNA-mediated silencing in flies but the data obtained by RNAi knockdown and with their new *gw182*-null alleles only show the effect with their new GW182-independent miRNA reporter. Therefore, it will be important for this study to show that a GW182-dependent reporter is sensitive to, at least, the loss of *gw182* using this new CRISPR/Cas9 KO alleles.

We fully agree with the Referee that it would be ideal to show that a GW182-dependent reporter is sensitive to the loss of *gw182* using our new CRISPR/Cas9 KO alleles. However, our *gw182* null mutant flies are lethal at the 2nd instar larvae stage, making it impossible for us to analyze the only previously described GW182-dependent reporter (McCann et al., PNAS 2011); this reporter was investigated through RNAi-mediated knockdown of *gw182* in the wing disc of 3rd

instar larvae, a stage our null mutant flies do not reach to survive. Moreover, this previous reporter lacked any internal control. We have thus spent more than a year to test various reporters and genetic crosses, searching for a GW182-dependent reporter that is functional in the 1st instar larvae before *gw182* null flies die. However, many of the previously described “miRNA sensor” reporters in flies were perfectly complementary to miRNAs (e.g., Brennecke et al., Cell 2003; Friggi-Grelin et al., Genetics 2008), which could potentially be silenced by the weak but potent target RNA cleavage activity of Ago1, making them inappropriate for monitoring the contribution of GW182. We have therefore performed our internally controlled GFP-mCherry dual reporter assay, which we originally performed in the miR-8-overexpressing eye disc (using *GMR-Gal4*; Fig. 1–2), now in the wing disc (using *ptc-Gal4*) of the 3rd instar larvae that express miR-8 endogenously. We found that RNAi of *gw182* modestly but significantly de-repressed the GFP fluorescence (Fig. R1, below), suggesting that this same reporter is silenced in a GW182-dependent manner in the eye disc (Fig. R1A) but in a GW182-independent manner in the wing disc (Fig. 2). However, as a control, de-repression by RNAi of *ago1* was only slight and statistically insignificant (Fig. R1B), probably because of inefficient RNAi of *ago1* in the wing disc. Unfortunately, it is extremely challenging to perform tissue dissection without contamination of surrounding wild-type tissues to quantify the efficiency of tissue-specific RNAi in these tiny discs at the 3rd instar larvae stage. We have thus decided not to include this data in our revised manuscript but instead carefully discuss the possibility of tissue-specific GW182 dependency. We hope the Referee will understand the technical difficulty regarding this point.

Fig. R1

Endogenous miR-8 is expressed in the wing disc pouch enclosed by white dotted line.

-As nicely shown by this group before, while both Argonautes can bind miRNAs and repress mRNA targets in flies, GW182 interacts only with Ago1 but not Ago2. Thus, it will be relevant here to test and compare how Ago2 vs Ago1 contributes to the repression of their new GW182-independent miR-8 reporter. That will help gather some mechanistic insights about how the GW182-independent silencing mechanism works in vivo.

The miR-8/miR-8* duplex contains multiple mismatches at positions 5, 11, and 15 and a G:U wobble at position 9, therefore it is expected to be preferentially loaded into Ago1 rather than Ago2 according to the “soring rule” in flies (Förstemann et al., Cell 2007; Tomari et al., Cell 2007). Indeed, RNAi of *ago2* in the eye disc did not cause any apparent de-repression of our reporters, as shown below. We have included this data as Figure S2 in our revised manuscript.

-The authors should discuss the potential mechanism involved in the GW182-independent repression seen with the miR-8 reporter, which appears different from those observed with miR-8 and miR-14 predicted targets. In the absence of gw182, the mRNA level of the miR-8 reporter is not affected (Figures 2D and 4C), while the level of putative miR-8 and miR-14 targets in larvae at the first instar stage is significantly increased in gw182-null animals (Figure 5).

We thank the Referee for raising this important point. We believe that above-discussed tissue-specific or developmental stage-specific GW182 dependency can explain the apparent difference between our miR-8 reporter and the miR-8 and miR-14 predicted targets, at least to some extent. This model also agrees with the previous interesting findings in *C. elegans*, where somatic miRNAs are dependent on GW182 but germline miRNAs function through another factor, GLH-1 (Dallaire et al., Dev Cell 2018). However, as described above, the data we have obtained so far are not strong enough to conclude this tissue-dependency model in flies, and we have carefully discussed this point in the revised manuscript. In addition to the tissue specificity, the sequences and structures of the 3' UTR regions and their associated RNA-binding proteins may also affect the GW182 dependency. Moreover, it was recently reported that more actively translated mRNAs are more sensitive to miRNA-mediated silencing (Kobayashi and Singer, Nat Commun. 2022), which may also influence the GW182 dependency. We are eager to explore these interesting possibilities in our future studies.

-It is also unclear why the miR-8 reporter mRNA level is increased in the absence of Ago1. The current model suggests that GW182 recruits CCR4-Not complex to induce degradation of microRNA targets. So why the silencing of this GW182-independent reporter will still be

affected at the mRNA level in Ago1-null animals?

We believe that, even in the absence of active recruitment of the CCR4-NOT complex, translationally repressed mRNAs can be less protected by ribosomes and thus more prone to degradation. Alternatively, but not mutually exclusively, Ago1 may recruit the mRNA decay machinery independently of GW182 in vivo, given that virtually all the previous mechanistic studies of miRNA-mediated silencing in flies were performed in cultured S2 cells or cell-free systems.

-Is there a reason for creating new nucleotides pairing after the seed region in the mutated miR-8 sensor? I am concerned that this new base-pairing can make this reporter now sensitive to Ago2 slicing activity, which is likely not the case for the non-mutated sensor.

We appreciate the Referee's concern. As discussed above, miR-8/miR-8* duplex is preferentially sorted into Ago1 in flies, and we did not detect any apparent changes in the expression of both the Sensor and Mutated reporters by *ago2* knockdown (new Figure S2). Of course, Ago1 also has a weak target cleavage activity, which can be effective for perfectly complementary targets, but it is likely to be negligible when the seed sequence is mismatched.

-Please proof edit the figure legends. Some typos and mistakes can be found.

Thank you for pointing out our errors. We have corrected them.

Referee #2:

Matsuura-Suzuki and Tomari present data examining the requirement of GW182 for miRNA-mediated gene regulation and larval development in *Drosophila*. The manuscript is clear, thoughtful, and very well crafted. Major findings and contributions include:

- creation of a well-controlled in vivo reporter for miR-mediated repression
- evidence that AGO1-RISC can effectively silence miRNA targets in fly eye discs even when GW182 levels are reduced
- generation of two GW182-null *Drosophila* alleles
- GW182-null flies can survive embryogenesis and first instar larval development
- GW182 is essential for development at an early second instar stage

- miR-8 can silence the in vivo reporter in GW182-null first instar larvae
- derepression of miR-8 and miR-14 targets in GW182-null larvae
- GW182-null larvae suffer misregulation of chitin-related genes and trachea system defects

The study is exciting and timely as several groups have reported GW182-independent silencing in vitro in recent years and yet, apart from the single study in *C. elegans* (PMID: 27935964), in vivo evidence is lacking. The study has the potential for a high impact on the miRNA field as it may direct researchers to consider mechanisms and functions for miRNAs outside of GW182, and may help mitigate against the tenuous, but sometimes accepted, assumption that all miRNAs function through GW182 family proteins (i.e. PMID: 34463618).

Because GW182 null flies do not develop to maturity, it is not possible to generate a stable homozygous knock-out line. Therefore, the major conclusions of the study hinge on the validity of the notion that miRNA function in the GW182-null larvae is not dependent on small, but functional, materially-deposited GW182 or gw182 mRNA reserves. Major concerns/suggestions:

1) Page 6: "Indeed, the expression of GW182 protein was at an undetectable level in the first instar larvae of gw null-1 and gw null-2 (Figure 3E)". Staring at Fig. 3E with my eyes slightly defocused, I believe I can detect a faint band around the size of GW182 in the null-1 lane and especially the null-2 lane. What does an over-exposed image of this blot look like? Can the authors quantitate band intensities to associate a numerical value with the limit of detection for GW182 protein levels?

We thank the Reviewer for raising this important point. We have made many attempts to accurately quantify the GW182 protein level in our null flies. However, unfortunately, the specific activity of our anti-GW182 antibody seems to have markedly decreased over the years, and we could now only detect faint GW182 bands (*) even in the wild-type 1st instar larvae samples, in contrast to the S2 cell lysate sample where GW182 bands are more discernible (we assume that the doublet reflects phosphorylation as reported in humans [Munakata et al., Genes 2021]).

Fig. R2

The over-exposed image of Fig. 3E looks like below, which we have now included in the revised manuscript. We fully agree with the Referee that there is a small but detectable amount of GW182 in the 1st instar larvae of *gw182* null flies, as a residual of the maternal deposition. We have therefore weakened our statement in the revised manuscript accordingly (please see point #5 below for the actual revised text).

2) An obvious orthogonal experiment would involve the creation of flies with mutations in the tryptophan-binding pockets in Ago1, as in PMID: 27935964. In this case, the Ago1 K785S + R813S double mutant would be expected to be impaired in binding GW motif I, which appears to be the primary Ago1-binding site in GW182 (PMID: 19383769, Fig. 1). Alternatively, a motif I GW182 mutant (i.e. W11A + W22A) is expected to be impaired in Ago1-binding. Have the authors considered experiments along these lines to support their conclusions regarding GW182-independent miRNA function?

We are grateful to the Referee for this constructive suggestion. We agree that the Ago1 K785S + R813S double mutant and/or the motif I GW182 mutant are extremely powerful tools.

However, creation of these flies requires precise genome editing or introduction of new transgenes driven by endogenous promoters, followed by multiple genetic crosses. Although we are eager to pursue these promising directions in the future, we would like to focus on the generation of *gw182* null flies and their analyses in our current manuscript.

3) Page 6: "These results suggest that AGO1-RISC can silence miRNA targets independently of GW182 in the fly eye disc." This suggestion seems rather strong considering that the knock-down of *gw182* mRNA by the inverted repeats is only ~2-fold (Fig. S1). How does this compare to the knock-down *ago1* in the *ago1-IR* line? Is *ago1* knock-down similar?

We appreciate the Referee's concern. In general, precisely evaluating the efficiency of tissue-specific RNAi (using GMR-Gal4) in the eye disc of the 3rd instar larvae is extremely challenging, because the dissection of this tiny tissue could inevitably be contaminated with surrounding, wild-type tissues. Indeed, the RT-qPCR result of *ago1* knockdown looks like below.

However, we reproducibly observe the typical "rough eye" phenotype associated with miRNA dysfunction in the adult flies after knockdown of *ago1* or *gw182* in the eye disc, suggesting that the knockdown is functionally effective in our experiments. We have included these data as Figure S1B and C and clarified the technical limitation of tissue-specific RT-qPCR in our manuscript.

Minor comments/suggestions:

4) Page 3: "The N-terminal glycine/tryptophan (GW) repeats of GW182 directly binds the two hydrophobic pockets (tryptophane-binding pockets) in the PIWI domain of AGO proteins (Elkayam et al., 2017; Schirle et al., 2014)" A third AGO protein tryptophan-binding pocket that directly binds GW repeats in TNRC6B was reported in PMID: 29576456.

We thank the Reviewer for raising this critical point and apologize for the inaccuracy. We have corrected this point in the revised manuscript.

5) Page 6: "It has been demonstrated that maternally deposited GW182 is depleted by 60-70 min after embryo laying (Schneider et al., 2006)." I am not sure this statement accurately reflects the data in Fig. 4 of Schneider et al. It looks to me that 60-70 min after egg deposition GW182 protein levels are about 50% of the level 0-10 min after egg deposition (Schneider et al., Fig. 4B, lower panel). GW182 levels then rise, which the authors attribute to zygotic transcription, making the fate of maternally deposited GW182 after the 60-70 min window unknown. Are these the data to which the authors are referring, or am I missing something here?

We initially followed the original authors' statement in the paper "The presumptive maternal GW contribution to the embryo appears to be depleted by 60–70 min AED followed by an increase in GW levels starting at 80 min AED (Fig. 4 B). The activation of zygotic *gw* transcription was confirmed by Northern blotting." However, we agree with the Referee that the fate of maternally deposited GW182 after the 60–70 min is formally unknown, and we apologize for any confusion. In conjunction with the re-evaluation of the residual amount of GW182 in the 1st instar larvae of null flies (please see above), we have weakened our statement as follows in the revised manuscript.

“It has been demonstrated that maternally deposited GW182 starts to decline at 60–70 min after embryo laying (Schneider et al., 2006). Although the fate of maternally deposited GW182 after 60–70 min is unknown, the time point of death of *gw182*-null flies is more than 48 hours after this time window. In the surviving first instar larvae of *gw^{null-1}* and *gw^{null-2}*, only trace amounts of GW182 protein were detectable (Figure 3E). These results suggest that flies can survive embryogenesis and first instar larval development even after maternal GW182 is mostly depleted, but GW182 becomes essential for development at an early second instar stage.”

“Thus, we concluded that miR-8 can silence the *in vivo* reporter independently of GW182 in first instar larvae, although it is formally possible that the residual amount of GW182 in *gw^{null-2}* is sufficient to support the full silencing activity.”

6) It might be helpful to display Fig. S3 alongside data in Fig. 6A,C (this is only a suggestion).

We thank the Referee for his/her kind suggestion. We have now included the original Fig. S3 as Fig. 6E in the revised manuscript.

Referee #3:

In the manuscript "The requirement of GW182 in miRNA-mediated gene silencing in *Drosophila* larval development," Suzuki and Tomari re-evaluate the previously addressed role of GW182 in miRNA-mediated gene regulation during *Drosophila* development. Based on two newly generated CRISPR-Cas9 genome editing-derived *gw182* null alleles, they report that flies depleted of GW182 survive until an early second instar larval stage. This is in contrast to previous findings employing a truncation mutation in the *gw182* locus, which indicated a requirement to enter gastrulation. Furthermore, GW182 appears to be dispensable for miRNA-mediated gene repression, since *gw182* null flies were capable of repressing a heterologous reporter construct. At the same time, targets of abundant miRNAs in fly larvae are stabilized in the absence of GW182 confirming its contribution to reach a full extent or robustness of miRNA-mediated mRNA repression. Finally, characterization of the steady-state transcriptome in first instar larvae suggests a downregulation of chitin-related genes and air filling defects of trachea upon depletion of GW182, demonstrating the importance of GW182 in trachea formation. Overall, the study addresses an important genetic conundrum in the genetic dependencies of GW182 in various model systems and implies a specific (possibly

miRNA-independent) function in larval development. I therefore recommend publication of the manuscript if the following comments/concerns can be adequately addressed.

Comments:

- RNAseq analyses compare transcript abundances between *gw182* null mutant flies and *yw* control flies. Hence, it is unclear to what extent the observed changes (i.e. upregulation of immune system-related transcripts and downregulation of chitin-related genes) are caused by loss-of-GW182 or different genetic background and/or stock-effects. I would strongly recommend to include a heterozygous sibling control in those experiments to distinguish these possibilities.

We thank the Referee for raising this important point of heterozygous sibling control. We have now performed RNA-seq of *gw182*^{-/-} homozygous flies and their *gw182*^{-/+} heterozygous and *gw182*^{+/+} homozygous siblings. As shown below, we observed significant upregulation of predicted targets of miR-8 and miR-14 in *gw182*^{-/-} flies compared to either *gw182*^{-/+} or *gw182*^{+/+} siblings, just like our original comparison between *gw182*^{-/-} flies and *yw* control flies. We have included these data as Fig. S5 in the revised manuscript.

- It would be helpful to comment on the relationship between upregulated mRNAs and the presence/absence of target sites for highly abundant miRNAs to get an impression on the relationship of transcriptome changes and miRNA-related GW182-biology.

As shown below, among all the upregulated transcripts in *gw^{null-2}* flies, predicted targets for miR-8 and miR-14 and those for the top 10 miRNAs accounted for ~10–20% and ~20–30%, respectively, compared to either *yw* flies, *-/+* siblings, or *+/+* siblings. We have included these data as Figure 5B and S5B in the revised manuscript.

gw^{null-2} vs. *yw*

gw^{null-2} -/- vs. -/+

gw^{null-2} -/- vs. +/+

- It is interesting that miR-8 and miR-14 are significantly upregulated in *gw182* null mutants. While those changes seem statistically significant, the authors refrain from including a possible explanation. Like mentioned above, this may be caused by genetic background, which can be addressed by involving comparisons to heterozygous siblings. Alternative explanations are certainly possible.

We have compared the expression levels of miR-8 and miR-14 in *gw182*^{-/-} homozygous flies and their *gw182*^{-/+} and *gw182*^{+/+} siblings by qRT-PCR. As shown below and in Fig. S5C in the revised manuscript, we indeed observed a tendency of upregulation of miR-8 and miR-14 by the absence of *gw182*, as in original Fig. 5B (now Fig. 5C). Nevertheless, predicted target mRNAs of miR-8 and miR-14 were significantly de-silenced (rather than over-silenced) in *gw182*^{-/-} homozygous flies compared to their *gw182*^{-/+} and *gw182*^{+/+} siblings, as shown above. We thus concluded that the observed changes in the RNAseq analyses are indeed caused by loss-of-GW182 rather than genetic background. We thank the Reviewer again for his/her constructive suggestion to include heterozygous siblings as an important control.

- Precise p-Values of changes in protein or RNA expression (etc.) that are deemed significant and non-significant should be reported to allow for unbiased interpretation.

- Figure 3d: n-values should be indicated to understand the population size that is underlying those data.

We appreciate the Referee's advice. We have added the p-values of changes and n-values in corresponding Figures and/or Figure Legends.

- Consider correcting in the first paragraph of discussion: "We found that, even in the absence of zygotic expression GW182, flies can complete the embryogenesis and develop into second instar larvae."

We apologize for any confusion. We have changed it simply to "We found that *gw182*-null flies can complete the embryogenesis and develop into second instar larvae."

Dear Dr Tomari,

Thank you for submitting a revised version of your manuscript. We find that you have addressed all the additional remarks raised by the referee. There remain only a few mainly editorial points that have to be addressed before I can extend formal acceptance of the manuscript:

1. DATA AVAILABILITY SECTION: in, but there is a typo - "Dava Availability"
2. COI: title needs renaming to "DISCLOSURE AND COMPETING INTERESTS STATEMENT"; In the "Disclosure and competing interests statement", please add the following disclaimer: "Yukihide Tomari is a member of the Advisory Editorial Board of The EMBO Journal. This has no bearing on the editorial consideration of this article for publication."
3. CRediT has replaced the traditional author contributions section because it offers a systematic, machine-readable author contributions format that allows for more effective research assessment. Please remove the Authors Contributions from the manuscript and use the free text boxes beneath each contributing author's name in our online submission system to add specific details on the author's contribution. More information is available in our guide to authors.
4. There is a reference to "data not shown" on page 8, According to our policy, which does not permit references to "data not shown", please include this information in the Appendix. Please see also <https://www.embopress.org/page/journal/14602075/authorguide#unpublisheddata>
5. Main and EV figures should be uploaded individually as high-resolution figure files; Figures S1-S5 should be renamed to Figure EV1-EV5 with the corresponding callouts
6. Appendix file needs to be in PDF format
7. Papers published in The EMBO Journal are accompanied online by a 'Synopsis' to enhance discoverability of the manuscript. It consists of A) a short (1-2 sentences) summary of the findings and their significance, B) 3-4 bullet points highlighting key results and C) a synopsis image that is 550x300-600 pixels large (width x height, jpeg or png format). You can either show a model or key data in the synopsis image. Please note that the image size is rather small and that text needs to be readable at the final size. Please send us this information together with the revised manuscript.
8. "Please note that the specific URLs for GSE192811 and GSE267756 datasets are not provided in the data availability statement."
9. Please note that the exact p values are not provided in the legends of figures 2c-d; 4b-c, supplementary figures 2b; 3b; 5a.
10. Please note that the measure of center for the error bars needs to be defined in the legends of figures 1e; 2c-d; 4b-c; 5c, supplementary figures 1a-b; 2b-c; 3b; 5c.
11. Please note that scale bar and its definition are missing for figures 1c; 2a-b; 4a, supplementary figures 2a; 3a.
12. Please change the section order to: title page with complete author information, abstract, keywords, introduction, results, discussion, materials & methods, data availability section, acknowledgements, disclosure and competing interests statement, references, main figure legends, tables, expanded figure legends.
13. Please provide source data. My colleague Hannah Sonntag will contact you seperately with more information.

With best regards,

Cornelius

Cornelius Schneider, PhD
Editor | The EMBO Journal
c.schneider@embojournal.org

When assembling figures, please refer to our figure preparation guideline in order to ensure proper formatting and readability in

print as well as on screen:

We realize that it is difficult to revise to a specific deadline. In the interest of protecting the conceptual advance provided by the work, we recommend a revision within 3 months (23rd Sep 2024). Please discuss the revision progress ahead of this time with the editor if you require more time to complete the revisions. Use the link below to submit your revision:

Referee #1:

The authors did an excellent job of adequately answering my concerns in this revised manuscript. I am supportive of the publication of this interesting and important study for the microRNA field.

Referee #2:

The authors have addressed all of my previous concerns with thought and care. I congratulate them on brining this study to fruition.

Referee #3:

Matsuura-Suzuki and Tomari present a comprehensive study on the role of GW182 in miRNA-mediated gene regulation during *Drosophila* development. The manuscript reports on the generation of two new CRISPR-Cas9 derived gw182 null alleles and their characterization. Key findings include the survival of gw182-null flies until the early second instar larval stage, the ability of these flies to repress a heterologous reporter construct independently of GW182, and the identification of specific developmental defects and gene expression changes in the absence of GW182.

The study addresses a significant gap in the understanding of miRNA function, particularly the extent to which miRNA-mediated gene silencing depends on GW182 in vivo. The findings challenge the previously accepted notion that GW182 is universally required for miRNA function and suggest that some miRNAs can function independently of GW182, at least in certain developmental contexts. This work has the potential to impact the broader miRNA research field by prompting further investigation into alternative mechanisms of miRNA action.

The manuscript was resubmitted after a longer revision time, owed to the pandemic and the career transition of the first author. The authors have addressed all major concerns raised in the review of the original version of the manuscript. I therefore support the publication of the revised version.

All editorial and formatting issues were resolved by the authors.

Dear Prof. Tomari,

I am pleased to inform you that your manuscript has been accepted for publication in the EMBO Journal.

Yours sincerely,

Cornelius Schneider, PhD
Editor
The EMBO Journal
c.schneider@embojournal.org
